



# NortheastChinaSoybeanYield20m: an annual soybean yield dataset at 20 m in Northeast China from 2019 to 2023

Jingyuan Xu[1,2], Xin Du[1,2], Taifeng Dong[3], Qiangzi Li[1,2], Yuan Zhang[1,2], Hongyan Wang[1,2], Jing Xiao[1,2], Jiashu Zhang[1,4], Yunqi Shen[1,2], Yong Dong[1,2]

[1]Aerospace Information Research Institute, Chinese Academy of Sciences, Beijing 100094, China
[2]University of Chinese Academy of Sciences, Beijing 100190, China
[3]Ottawa Research and Development Centre, Agriculture and Agri-Food Canada, 960 Carling Ave, Ottawa, ON K1A 0C6, Canada
[4]School of Science, China University of Geosciences (Beijing), Beijing 100083, China

*Correspondence to*: Xin Du (duxin@aircas.ac.cn)

**Abstract.** Accurate monitoring of crop yield is important for ensuring food security. However, exiting yield datasets with a coarse spatial resolution are inadequate for capturing small scale spatial heterogeneity. Current yield estimation methods, such as machine learning models or the assimilation of remotely sensed biophysical variables into crop growth models, depend heavily on ground observations and involve significant computational costs. To solve these problems, a hybrid framework coupling the World Food Studies Simulation Model (WOFOST) and the Gated Recurrent Unit model (GRU) was proposed to generate a 20 m soybean yield dataset in Northeast China from 2019 to 2023 (NortheastChindaSoybeanYield20m). A soybean growth dataset was first generated based on the WOFOST that simulated various production scenarios (climates, crop varieties, soil types and agro-managements). The GRU model was then trained for characterizing relationships between model simulated LAI and soybean yield. The trained model was then applied for soybean yield estimation in Northeast China using time series LAI of different growth stages derived from Sentinel-2. The accuracy of the dataset was evaluated by in-situ measured and statistical data. The overall accuracy was 287.44 kg ha$^{-1}$ and 272.36 kg ha$^{-1}$ in the root mean squared error (RMSE) for field and regional scale, respectively. Stable results were achieved through the years with mean relative error (MRE) on average of 11.46 % in municipal scale and 7.94 % in provincial scale. Results demonstrated that the model was able to capture spatial-temporal variation of soybean yield. The NortheastChinaSoybeanYield20m was able to capture spatial-temporal variation of soybean yield, which can be applied for optimizing soybean production distribution and guiding agricultural decision-making. The NortheastChinaSoybeanYield20m dataset can be downloaded from https://doi.org/10.5281/zenodo.14263103 (Xu et al., 2024)





## 1 Introduction

Soybean is an important crop for both food and oil production, supplying more than a quarter of the world's edible protein (Graham and Vance, 2003). With rapid population growth, the demand for food continues to rise. It is projected that global demand for soybeans will increase by 46 % by 2050 (Falcon et al., 2022). As an important traded commodity, the impact of soybean production in major producing countries can be felt globally through international trade (Qiao et al., 2023) and thus affecting agricultural economies around the world. China is the world's largest consumer of soybeans (FAOSTAT, 2022). The

soybean demand in China is heavily dependent on international trade (Zhao et al., 2023). In essence, accurate monitor of soybean yield is vital for fostering sustainable agriculture, ensuring food security, and maintaining economic stability on a global scale. In addition, it is also beneficial for farmers for managing field practices, supporting agricultural insurance and improving poverty alleviation efforts (Zhuo et al., 2022).

Remote sensing data has provided time series data support for crop yield estimation at different scales (e.g., field, region

and country) (Dong et al., 2020; Hunt et al., 2019; Y. Zhao et al., 2023). The methodologies for crop yield estimation based on remote sensing data can be further divided into data-driven methods and knowledge-driven methods.

Data-driven methods can explore the linear or nonlinear relationships between satellite-derived variables and crop yield (Tennakoon et al., 1992). Variables retrieved from remote sensing data include leaf area index (LAI), fraction of absorbed photosynthetically active radiation (FAPAR), leaf chlorophyll content (LCC) and vegetation indexes (VIs) (Ang et al., 2022;

Xie et al., 2019). Due to their capability in handling large volumes of input data and nonlinear tasks, machine learning algorithms such as Random Forest (RF), Artificial Neural Networks (ANN) have been widely applied in crop yield estimations (Pang et al., 2022; Tian et al., 2021; Yildirim et al., 2022). In reality, it is difficult to meet the demand of high-precision yield estimation using a single type of indicator. Some literatures combined soil moisture (SM), evapotranspiration (ET) with crop physiological parameters to participate in modeling (Islam et al., 2023; Ji et al., 2022). These methods can extract effective

information from a large amount of structured/unstructured data without manual intervention to meet the needs of multi-source data. However, one disadvantage of data-driven approaches is that they dependent heavily on large training datasets (Cao et al., 2021). It is often challenging to obtain sufficient ground samples in large area application. Outliers and noise of the obtained data can also degrade the performance of the model and increase uncertainty in the prediction (Taylor et al., 2007). In addition, due to the lack of theoretical support, data-driven models cannot explain the causal relationship between input features and

predicted outputs which results in poor spatial-temporal generalization capability (Gevaert, 2022).

Different from data-driven methods, knowledge-driven crop growth models can characterize the entire evolution of crops from sowing to harvesting (Huang et al., 2019). These models were built on the basis of agronomic mechanism knowledge. The growth of crops was simulated by combining environmental factors (e.g., climate conditions and soil characteristics) with crop growth processes (e.g., photosynthesis, respiration and transpiration) (Gaso et al., 2024). According to knowledge-driven

types, crop growth models can be further categorized into light use efficiency models (e.g., SAFY (Duchemin et al., 2008)), soil-driven models (e.g., AquaCrop (Steduto et al., 2009)), atmospheric-driven models (e.g., WOFOST (Diepen et al., 1989))





and so on. However, the application of crop growth models in large area is often constrained by the lack of available model input data which varies with space and time (Dokoohaki et al., 2021). The uncertainty of model parameters will further increase the bias in crop yield estimation. Since remote sensing data can provide spatial inputs for crop growth models, data assimilation methods that combining spatial-temporal monitoring of remote sensing data with the simulation of crop growth model have been developed (Huang et al., 2024). However, the high spatial resolution of remote sensing data significantly increases the computational cost of the data assimilation process (Huang et al., 2019). As a result, crop growth models have been hampered in large area applications.

Given the difficulties above, integrating the strengths of data-driven and knowledge-driven models to enhance spatial-temporal generalization capabilities of the model and to address sparse training sample issues becomes a critical focus in crop yield estimations. A hybrid method coupling crop growth model with machine learning algorithm has raising increasing interest (Ren et al., 2023; Xie and Huang, 2021). The system utilized simulated results from crop growth models as input for machine learning. Input features of the predicted model include meteorological factors (Isia et al., 2022), soil characteristic factors (Saravi et al., 2020), crop growth factors (Paudel et al., 2021), management factors (Ren et al., 2023) and observation geometry for remote sensing data (Chen et al., 2022). Many studies have demonstrated the ability of hybrid methods in crop yield estimation (Feng et al., 2020; Xie and Huang, 2021; Yang et al., 2021). On the one hand, the application of crop growth model can provide biophysical constraints for machine learning modeling and provide a sufficient simulation dataset for training the machine learning model. On the other hand, the combination of machine learning improves the computational efficiency of yield estimation at the regional scale compared with data assimilation method (Xie and Huang, 2021). However, the extraction of characteristic factors was generally based on the entire growth stage of the crop in existing studies (Pinke and Lövei, 2017; Wang et al., 2015). This increased the cost of model calculation and might not capture the influence of characteristic factors to a specific stage. Several studies obtained characteristic factors on a monthly scale or based on field observation dates (Everingham et al., 2016; Kern et al., 2018). However, these studies did not consider the specific growth stages of crops. As for the model use, the deep learning models, such as Long Short-Term Memory (LSTM) and GRU model, have a better ability to capture time series information but have not been widely used in hybrid modeling. According to surveys, there is currently no high-resolution soybean yield dataset available in the main production regions of China for studying spatiotemporal patterns of soybean production. Therefore, it is urgent to leverage the advantages of hybrid modeling method to create a high-resolution soybean yield dataset to further guide agricultural practices.

This study developed a hybrid model coupling data-driven and knowledge-driven models for estimation of soybean yield in Northeast China. The WOFOST model was first adopted to simulate a multi-scenario soybean growth dataset to train the GRU model. The time series Sentinel-2 data of different soybean growth stages were then input into the GRU model to estimate soybean yield. Our study aims to address the following objectives: (1) Designing a hybrid model coupling crop growth model and deep learning model for soybean yield estimation; (2) Generating a high resolution (20 m) soybean yield dataset in Northeast China (NortheastChinaSoybeanYield20m) from 2019 to 2023; (3) Exploring the accuracy of the datasetat multi-scale applications.





## 2 Data preparation and preprocessing

### 2.1 Study areas

The study was carried out in Northeast China (38°40′ N to 53°34′ N, 115°05′ E to 135°02′ E) covering Heilongjiang, Jilin, Liaoning province, as well as the eastern parts of the Inner Mongolia Autonomous Region (IMAR) (Fig. 1). It comprises 40 cities and its total area is approximately 1.24 million km². The region is characterized by continental monsoon climate. The annual accumulated temperature (≥ 10 °C) ranges from 2200 to 3600 °C (Pu et al., 2019), and the frost-free period is between 140 to 170 days (Tan et al., 2014). The average annual precipitation decreases from east (1000 mm) to west (350 mm) (Zhao et al., 2011). The main soil types in the study area include brown coniferous forest soil, dark brown forest soil, forest steppe chernozem and meadow grassland chernozem soil (Pu et al., 2019). Soybean is one of the three main crops in the study area. It is mainly cultivated in the northern parts of Songliao plain rotated with maize. Around 97 % of the soybean is rainfed (Guo et al., 2022; Yu et al., 2020). The growth period generally lasts from May to late September (Zhao et al., 2021). The soybean yield accounts for 64 % of the total yearly soybean yield in China (National Bureau of Statistics of China (NBSC), 2023).

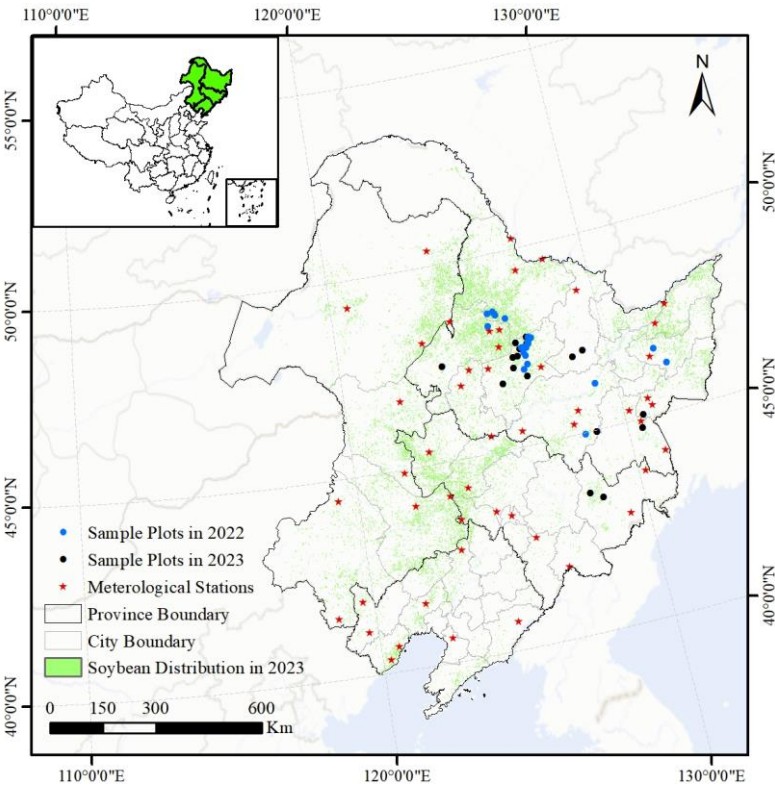

**Figure 1: Location of the study area and the distribution of sample plots in two years (2022 and 2023) and selected meteorological stations.**



## 2.2 Data collections

### 2.2.1 In-situ measurement data

Field-scale yield data was separately collected through field investigation in September 2022 and 2023. In each year, a total of 21 and 18 sample plots were selected, respectively (Fig. 1). Within each sample plot that was around 100 m × 100 m in area, nine quadrats with area of 1 m × 1 m were selected randomly for destructive sampling of yield in soybean. The central location of each quadrat was recorded using a GPS device with accuracy of 1 m. The harvested beans were then oven-dried about 72 hours in Sihua Agricultural Research Institute, Hailun to determine the yield. Finally, the average yield for the selected nine quadrats represents the soybean yield of the sample plot.

Field measured LAI data of soybean was obtained from the Common application support platform for land observation satellite (CAPLOS, https://124.16.188.131:9699/web/server3/build/#/Guide). It provides in-situ measured biophysical variables (e.g., LAI and vegetation cover) from ground stations for the validation of authenticity and theoretical research on remote sensing product retrieval algorithms. LAI was measured using the LICOR LAI-2200. At each site, the instrument was positioned about 5 cm above the ground to get six readings of radiation transmitting by canopy and was positioned above the canopy to get a reading for incoming radiation. The data was cleaned to remove outliers, missing or duplicate values. Finally, a total of 94 observations of LAI for the three soybean growing seasons from 2021 to 2023 were collected.

### 2.2.2 Meteorological data

In this study, two different climate datasets were used.

The meteorological data used in this study came from the meteorological stations of the National Meteorological Information Center (http://data.cma.cn). There are 238 meteorological stations within the study area. Here 51 of the meteorological stations that located within 1 km buffer zone of the soybean cultivation areas were selected (Fig. 1). The meteorological datasets generally include insolation duration (h), minimum temperature (℃), maximum temperature (℃), daily average temperature (℃), average water vapor pressure (kPa), average wind speed (m sec$^{-1}$), precipitation (mm) and snow-depth (cm). Observed data from 1980 to 2021 of the 51 selected stations were collected. They were used as input climate parameters for the WOFOST model to drive simulations.

The spatiotemporal distribution of meteorological data was obtained from the ERA5-land Daily Aggregated - ECMWF Climate Reanalysis Product. The ERA5-land is a global climate reanalysis product that provides continuous climate data at a resolution of 0.1° × 0.1° (e.g., air temperature and atmospheric pressure) starting from 1950. It provides a daily aggregated air temperature data at 2 m above the surface of land measured in kelvin (K). All the selected data for the soybean growth periods from 2019 to 2023 were obtained from the Google Earth Engine (http://earthengine.google.com). The product was resampled to 20 m using bilinear interpolation model and was then used to calculate the soybean phenology for preperation of yield estimations.



### 2.2.3 Soil data

Soil data was obtained from 1:1000,000 Chinese soil database downloading from Geographic Data Sharing Infrastructure, global resources data cloud (www.gis5g.com). The dataset covers the distribution of various soil types and their main chemical characteristics across the country.

### 2.2.4 Satellite imagery data

Two satellite data include: 1) Sentinel-2 images (Level L2A Top-of-Atmosphere reflectance product), 2) the Moderate Resolution Imaging Spectroradiometer Product (MCD15A3H.061) were used in this study. All of the data collected in the four soybean growth periods (2019-2023) were downloaded from Google Earth Engine (http://earthengine.google.com).

The Multi-Spectral Instrument (MSI) on the Sentinel-2 satellites (including Sentinel-2A and Sentinel-2B) provides high spatial resolution imagery with resolutions of 10 m, 20 m and 60 m, and a temporal resolution of 5 days. Only cloud-free images were selected. The band with 60 m was excluded in the analysis, and the bands of 10 and 20 m were used. To match the red-edge bands, the four 10 m bands including B2 / B3 / B4 / B8 was resampled to 20 m using bilinear interpolation model.

The MCD15A3H (Collection 6.1, Level 4) provides 500 m LAI and FAPAR product with 4-day composite. The LAI product was chosen in this study. The optimal pixel from data of MODIS sensors on NASA's Terra and Aqua satellites within the 4 days is selected for inversion of LAI. The retrieval algorithm is based on a look-up-table (LUT) derived from a 3D radiative transfer model (Knyazikhin et al., 2018). It estimates LAI by using the ratio of red to near-infrared reflectance. An empirical model based on relationships between LAI and NDVI is served as an alternative choice when the main algorithm fails. As the product provides a higher temporal resolution than Sentinel-2 satellites, yield estimations were also done based on LAI product to better characterize the rapid changes in soybean growth stage. The MODIS yield maps were then applied for bias correction of Sentinel-2 yield maps.

### 2.2.5 Crop distribution data

The soybean distribution maps from 2019 to 2023 were obtained from Zhao et al., (2022). The study proposed the optimal identification feature (OIF) knowledge graph coupling with moment-preserving segmentation method for crop types identification through the crop growth season. The overall accuracy and the producer accuracy for maize, soybean and rice was higher than 90 % and 93 %, respectively.

### 2.2.6 Statistical data

The crop yield records (1980-2022) of the Statistical Yearbook provided by Statistic Bureau of Heilongjiang (http://tjj.hlj.gov.cn), Jilin (http://tjj.jl.gov.cn), Liaoning (https://tjj.ln.gov.cn) as well as IMAR (https://tj.nmg.gov.cn) were collected for the validation of crop yield estimation. For the statistical yield in 2022, the data covered a limited number of cities as the Statistical Yearbook in 2022 has not been fully released. The data from 1980-2022 were all used to validate the



reasonability of model simulations for construction of soybean growth dataset. For further yield estimates at regional scale, data from 2019 to 2022 were applied for accuracy evaluation.

## 3 Methodology

175     Our proposed hybrid model utilizes both the advantages of machine learning in data mining and the mechanism advantages of crop growth model. Figure 2 presents the flowchart of the hybrid methodology for soybean yield estimation. It mainly includes 1) Constructing multi-scenario soybean growth dataset, 2) Modeling for relationships between soybean yield and inputs including meteorological data, soil types, crop varieties and agro-managements, and 3) Estimating soybean yield in multi-scale from LAI derived from remote sensing.





180

**Figure 2: The flowchart of the overall yield estimation methodology in this study.**

## 3.1 Construction of multi-scenario soybean growth dataset

Soybean growth dataset was a multi-scenario dataset based on simulations of crop growth model. It was a quantitative

description of the formation process of soybean yield, considering the association between soybean yield and growth state as

185   well as environmental conditions. It was constructed by comprehensively and systematically simulating the crop growth under

various agricultural production scenarios such as different meteorological conditions, soil types, crop varieties and agro-

managements.



In this study, the World Food Studies Simulation Model (WOFOST) model (Diepen et al., 1989) was employed to generate knowledge-based soybean growth dataset. Detailed information about the model could be found in Huang et al., (2015). The Python Crop Simulation Environment (PCSE) framework version 5.5 was used in this study. The generic process description of the model was well-suited for conducting large-scale simulations (Zhuo et al., 2023). Based on the theory of $CO_2$, the WOFOST was developed to simulate daily changes of biomass and final grain yield for specific crop types. The crop growth process was regulated by crop development stages (DVS) in WOFOST with 0 indicating emergence, 1 indicating anthesis and 2 indicating maturity (Diepen et al., 1989). Through the crop growth simulation, the model can calculate and predict the crop yield.

### 3.1.1 Preparation of model input parameters

The WOFOST model employs a range of parameters including meteorology, soil, crop and agro-management for crop simulation (Diepen et al., 1989). None of advanced optimization was utilized in this study, since our objective is not to determine the globally optimal value of model parameters. The model parameters were set to accommodate various agricultural production scenarios during the soybean growth period without calibration with in-situ data. The values of the parameters were determined from four sources: ground observation, online database, literatures and the initial values provided in WOFOST.

(1) Meteorological parameters

The meteorological parameters required in WOFOST is shown in Table 1. To account for various meteorological conditions, the meteorological data collected from the selected 51 meteorological stations over a period of 42 years (1980-2021) was all utilized to provide values of these parameters. The meteorological inputs were then processed into the format recognized by the model.

**Table 1 Meteorological parameters required in WOFOST.**

| Parameter | Description | Units |
|-----------|-------------|-------|
| IRRAD | Incoming global shortware radiation | KJ m$^{-2}$ d$^{-1}$ |
| TMIN | Daily minimum temperature | °C |
| TMAX | Daily maximum temperature | °C |
| VAP | Daily mean vapour pressure | kPa |
| WIND | Daily mean windspeed at 2 m above the surface | m s$^{-1}$ |
| RAIN | Daily rainfall | mm |
| SNOWDEPTH | Snow depth | cm |

(2) Soil parameters

The soil parameters in the WOFOST mainly include soil moisture content at wilting point (SMW), field capacity (SMFCF) and saturation (SM0) as well as hydraulic conductivity of saturated soil (K0). In our study, they were acquired from the





1:1000,000 Chinese soil database. The soil texture in study area is predominantly loam soil. The loam soil is further divided into sandy loam, light loam, medium loam and heavy loam. The value settings of soil parameters for different soil types in the study area was presented in Table 2.

**Table 2 Values of main soil parameters in WOFOST.**

| Soil type | SMW (cm$^3$ cm$^{-3}$) | SMFCF (cm$^3$ cm$^{-3}$) | SM0 (cm$^3$ cm$^{-3}$) | K0 (cm d$^{-1}$) |
|---|---|---|---|---|
| Sandy loam | 0.060 | 0.280 | 0.350 | 22.6 |
| Light loam | 0.090 | 0.280 | 0.340 | 19.3 |
| Medium loam | 0.110 | 0.280 | 0.340 | 18.1 |
| Heavy loam | 0.194 | 0.355 | 0.356 | 34.6 |

(3) Crop-specific parameters

In this study, five different soybean varieties were considered in the study area to enhance the diversity of cultivars in the simulation, named early maturity, medium-early maturity, intermediate maturity, medium-late maturity and late maturity. They were designed for planting in the five thermal zones in Heilongjiang Province (Qu et al., 2023) (Table 3). To determine the distribution of soybean varieties in the study area, the ten thermal zones divided in Northeast China by Wang et al., (2022) were acquired. The thermal zones were calculated with a temperature difference of 200 ℃ d based on the thermal zones in Heilongjiang Province using historical meteorological data. The soybean varieties suitable for various thermal zones was then determined based on the demand of accumulated temperature (Table 3).

**Table 3 The division standard of the thermal zones and soybean varieties in Northeast China.**

| Thermal zones | Annual accumulated temperature ≥ 10 ℃ (℃ d) | Corresponding thermal zones in Heilongjiang Province | Soybean varieties |
|---|---|---|---|
| 1$^{st}$ thermal zone | ≥ 3500 | | |
| 2$^{nd}$ thermal zone | 3300 - 3500 | | |
| 3$^{rd}$ thermal zone | 3100 - 3300 | 1$^{st}$ thermal zone | late maturity |
| 4$^{th}$ thermal zone | 2900 - 3100 | | |
| 5$^{th}$ thermal zone | 2700 - 2900 | | |
| 6$^{th}$ thermal zone | 2500 - 2700 | 2$^{nd}$ thermal zone | medium-late maturity |
| 7$^{th}$ thermal zone | 2300 - 2500 | 3$^{rd}$ thermal zone | intermediate maturity |
| 8$^{th}$ thermal zone | 2100 - 2300 | 4$^{th}$ thermal zone | medium-early maturity |
| 9$^{th}$ thermal zone | 1900 - 2100 | 5$^{th}$ thermal zone | early maturity |
| 10$^{th}$ thermal zone | ≤ 1900 | | |

In the WOFOST model, soybean growth stages are mainly determined by temperature-related parameters including the
minimum and maximum threshold temperature for emergence (TBASEM, TEFFMX, respectively), accumulated temperature





($T_e$) from sowing to emergence (TSUMEM), from emergence to anthesis (TSUM1) and from anthesis to maturity (TSUM2). The accumulated temperature for different growth stage is sensitive to crop varieties according to the study of Qu et al., (2023). The values of main crop parameters for different soybean varieties were shown in Table 4. They were set according to the historical meteorological data and observation data of soybean and had been validated using actual development periods (Qu

et al., 2023). Other crop parameters used the default values of the WOFOST model or the optimal values from the study of Sun et al., (2022).

**Table 4 Values of main crop parameters in WOFOST.**

| Crop type | TBASEM (°C) | TEFFMX (°C) | TSUMEM (°C d) | TSUM1 (°C d) | TSUM2 (°C d) |
|---|---|---|---|---|---|
| Early maturity | 8 | 22 | 70 | 450 | 660 |
| Medium-early maturity | 8 | 22 | 70 | 480 | 770 |
| Intermediate maturity | 8 | 22 | 70 | 520 | 870 |
| Medium-late maturity | 8 | 22 | 70 | 540 | 960 |
| Late maturity | 8 | 22 | 70 | 580 | 1000 |

(4) Agro-management parameters

Soybean growing in the study area is mainly rainfed, and is received fertilizer management practices from the local

government. Therefore, the water-limited mode of the WOFOST was employed for soybean simulation. The water-limited model mainly requires the planting date for starting growth simulation. Soybeans in study area is typically sown between late April and late May and are seldom subjected to nutrient stress. Four planting dates 20 April, 30 April, 10 May, and 20 May to represent different agro-management scenarios, were set for model simulation.

**3.1.2 Multi-scenarios crop simulations**

After parameter preparation, a soybean growth dataset was constructed through model simulations which accounted for the multi-scenarios in agricultural production. The four different types of model parameters were arranged and combined to generate various simulation scenarios. The scenarios were then put into the model for simulation. Finally, a dataset containing more than 8,000 available simulations were generated.

**3.2 Development of the Grated Recurrent Unit model (GRU)**

The GRU was trained to estimate soybean yields in this study. The structure of a GRU cell is shown in Fig. 3. GRU controls the flow of information through update and reset gates (Cho et al., 2014). The update gate aims to control how much of the past information that are retrained and will be used in the future calculation. The reset gate aims to evaluate whether the remained previous information can be ignored in the new candidate hidden state. The use of two gates maintains the balance between retaining the hold hidden state and incorporating new information (Peng and Yili, 2022; Zhang et al., 2022). This





improves the training speed of the model and helps mitigate the vanishing gradient problem during training. Since GRU can effectively capture long-term dependencies in time series data, it has achieved good performance in applications of crop yield estimation (Gopi and Karthikeyan, 2023; Ren et al., 2023). The computation of a GRU unit can be summarized by the following equations:

$$R_t = \sigma(W_r \cdot [X_t, H_{t-1}] + b_r) \tag{1}$$

$$Z_t = \sigma(W_z \cdot [X_t, H_{t-1}] + b_z) \tag{2}$$

$$\widetilde{H_t} = \tanh(W_{\tilde{h}} \cdot X_t + W_{\tilde{h}} \cdot (R_t \odot H_{t-1}) + b_{\tilde{h}}) \tag{3}$$

$$H_t = (1 - Z_t) \odot H_{t-1} + Z_t \odot \widetilde{H_t} \tag{4}$$

where $R_t$ and $Z_t$ represents the activation vector of reset and update gates, respectively; $\widetilde{H_t}$ represents the potential update vector; $H_t$ and $X_t$ represent the hidden state output and the input at time t; $\sigma$ and tanh are the sigmoid function and the

hyperbolic tangent function; $W_r$, $W_z$ and $W_{\tilde{h}}$ represent the reset gate weight, the update gate weight and the update candidate weight, respectively; b is the bias vector of the parameters.

The GRU layer is connected with a fully connected layer. The output of the network at time t ($Y_t$) is finally determined by multiplying the hidden states of all cells in the GRU layer by the weights of the fully connected layer:

$$Y_t = W_y \cdot H_t + b_y \tag{5}$$

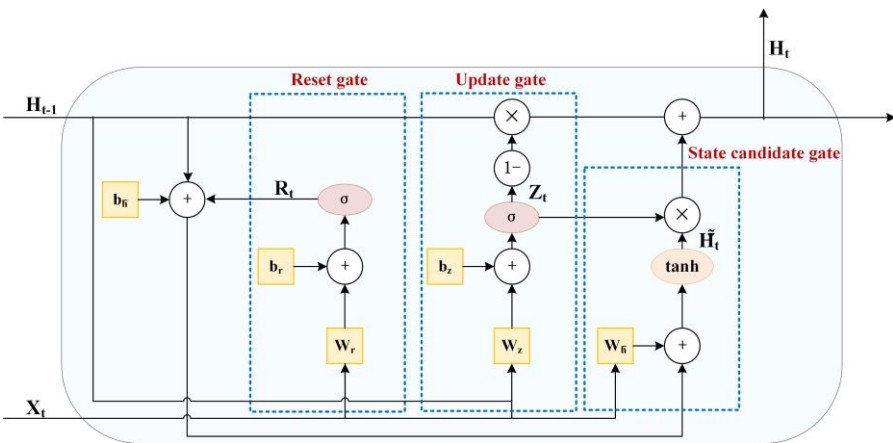


**Figure 3: Structure of a GRU cell.**

In this study, the GRU model was constructed based on TensorFlow 2.6. LAI was selected as the input feature of the model. As a crucial state variable in WOFOST, LAI signifies the photosynthetic capability of crops and can effectively characterize the potential yield (Huang et al., 2015). Accounting for the computational efficiency of the model in large areas,

the average value of LAI was used as the input feature of GRU. To better capture the growth dynamics of soybean, the mean LAI at vegetative (from emergence to flowering) and reproductive (from flowering to maturity) growth period in the soybean growth dataset were calculated, represented as LAI$_{mean1}$ and LAI$_{mean2}$. The two LAI values derived from the simulations, serving as temporal input features, were then combined with model simulated soybean yield in knowledge base to create the





simulated dataset for the GRU model. The simulated dataset was splits into training and testing datasets using 10-fold cross-validation. The hyperparameters of GRU was optimized using a grid search method (Açikkar, 2024). The root mean squared error (RMSE, Eq. (10)) was applied to assess the predictive performance of different set of hyperparameters. After optimization of each fold, the hyperparameters that yielded the smallest predictive error were selected as the optimal ones.

### 3.3 Generation of NortheastChinaSoybeanYield20m

### 3.3.1 Determination of soybean phenology

Due to the spatial variability of soybean phenology at the regional scale (Gaso et al., 2024), it is necessary to extract phenological stages of soybean before yield estimation. Based on the soybean varieties suitable for different thermal zones (Table 3) and the division of thermal zones in the study area, spatial distribution of soybean planting types in the study area was produced (Fig. A1). Daily aggregated air temperature data ERA5-land was applied for calculation of soybean phenology for different variaties regionally based on the $T_e$ (Eq. (6)):

$$T_e = \begin{cases} 0, (T_{mean} \leq T_{base}) \\ T_{mean} - T_{base}, (T_{base} < T_{mean} < T_{max}) \\ T_{max} - T_{base}, (T_{mean} \geq T_{max}) \end{cases} \tag{6}$$

where $T_{mean}$ represents the daily mean temperature, $T_{base}$ and $T_{max}$ represent the minimum and the maximum temperature for soybean development, respectively. Soybean growth proceeds from a growth stage to the next stage when $T_e$ reached the threshold required for growth. In this study, $T_{base}$ was set to 8 °C and $T_{max}$ was set to 37 °C (Allen et al., 1997; Choi et al., 2016).

Based on field surveys and literatures, the planting dates of soybean were uniformly set as 5 May for Heilongjiang Province and Inner Mongolia Autonomous Region, and 1 May for Jilin and Liaoning Province (Huang and Liu, 2024). The emergence date of soybean in Northeast China should not exceed 1 June at the latest (Mei et al., 2024). In addition, soybeans generally mature before 1 October (Huang and Liu, 2024). Under the constraints of $T_e$ (Table 2) and the agro-management, we calculated the emergence, anthesis and maturation time of soybeans in each Sentinel-2 pixel of planting areas from 2019 to 2023. Based on changes in agricultural phenology and the revisit cycle of remote sensing satellites, the phenological calculation results for each year were further clustered into 10 classes using K-means clustering method (Jain and Dubes, 1988). Remote sensing imagery were then obtained for different classes accordingly for yield estimation.

### 3.3.2 Model estimations of soybean yield

The estimation of soybean yield was mainly based on Sentinel-2 data. Red-edge normalized difference vegetation index (NDVI$_{RE}$) (Gitelson and Merzlyak, 1994)was utilized to derived the model input feature, LAI (Eq. (7)).

$$NDVI_{RE} = \frac{NIR - RE}{NIR + RE} \tag{7}$$

where NIR and RE represent the B8A and B5 band of Sentinel-2 data, respectively.





The LAI of soybean was characterized by a linear fit with $NDVI_{RE}$ (Eq. (8)). The linear fitting has been validated using a multi-crop dataset with $R^2$ of 0.732 and RMSE of 0.69 (Pasqualotto et al., 2019) which showed a unified potential for LAI

estimation in various crops.

$$LAI = 5.405 \cdot NDVI_{RE} - 0.114 \tag{8}$$

The average value of LAI for different growth stages of soybean ($LAI_{mean1}$ and $LAI_{mean2}$) was calculated. The two LAI was then input into the model for yield prediction. For pixels which had no observation either in vegetation or reproductive stage, LAI was replaced by the surrounding eight pixels instead. Finally, the yield maps with 20 m spatial resolution were

marked by soybean distribution maps to exclude non-soybean areas.

For large area estimations, a total of 194 Sentinel-2 tiles were required to fully cover the study area. Affected by cloud cover, the frequency of available data varied across each tile. Therefore, the yield maps often exhibited discontinuities along the edges of different tiles. This seaming effect could obscure real yield variations. To minimize the Sentinel-2 seaming effect, a bias correction method was employed following Azzari et al., (2017). The main idea of the bias correction approach should

be present in here. In this framework, MODIS imagery was used for intercalibration. Due to its higher temporal resolution and broader image coverage, MODIS generally provided more continuous estimation results. For correction, yield maps were also generated using MODIS LAI products. We utilized the estimation results from MODIS to calibrate the mean yield for each Sentinel-2 tile. The difference between the mean value of the yield derived from MODIS for the region cover the tile and the initial Sentinel-2 estimations was then added to the Sentinel-2 values. Through this correction, the seams within yield maps

were alleviated while the sub-tile variation of yields were preserved.

**3.4 Accuracy evaluation**

The accuracy of generated NortheastChinaSoybeanYield20m (2019-2023) was evaluated on multiple scales. For field scale, in-situ yield data in 2022 and 2023 was used for assessment. For regional scale, the mean soybean yield for each city and province were separately calculated for each year, and compared with the statistical data. Accuracy evaluation was based on

the coefficient of determination ($R^2$, Eq. (9)), the root mean squared error (RMSE, Eq. (10)) and mean relative error (MRE, Eq. (11)).\

$$R^2 = 1 - \frac{\sum_i (y_{o,i} - y_{m,i})^2}{\sum_i (y_{o,i} - \overline{y_o})^2} \tag{9}$$

$$RMSE = \sqrt{\frac{\sum_{i=1}^n (y_{o,i} - y_{m,i})^2}{n}} \tag{10}$$

$$MRE = \frac{\sum_{i=1}^n |y_{o,i} - y_{m,i}|}{n \cdot y_{o,i}} \tag{11}$$

where $y_{o,i}$ and $y_{m,i}$ represent the actual yield (observed or statistical yield) and model estimated yield, respectively, $\overline{y_0}$ is the mean value of the actual yield.



# 4 Results and analysis

## 4.1 Simulations of the WOFOST model

As the input feature, the accuracy of LAI simulated by WOFOST directly impacted the predictive capability of the GRU model
for soybean yield. Field-measured LAI from 2021 to 2023 was used to validate the reasonability of model simulated LAI.
5,000 simulated LAI curves were randomly selected and the mean value was calculated for comparison with ground
observation data (Fig. 4). The results indicated that the changes of simulated LAI were generally aligned with the observed
variations from field measurements. The range of simulated results encompassed more than 88 % of the field-measured sample
sites (n = 83), indicating a high level of confidence.

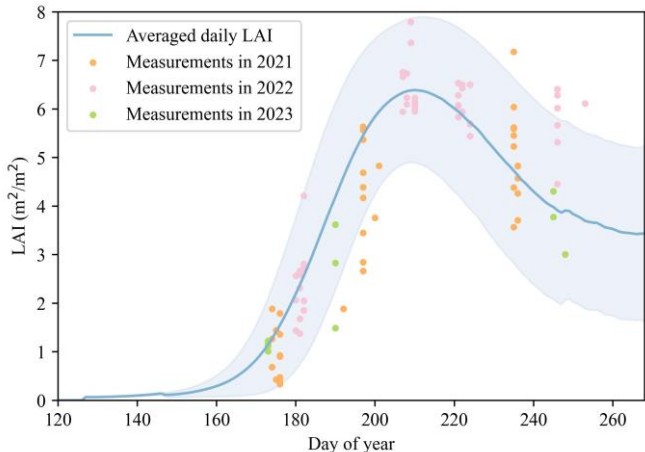


**Figure 4: Comparison of averaged daily LAI randomly selected from model simulations (n = 5000) with field-measured LAI in 2021
(n = 38), 2022 (n = 46) and 2023 (n = 10). The gray shading represents one standard deviation, indicating the uncertainty in LAI
simulation.**

Figure 5 displays the histogram distribution of simulated soybean yields. The result indicates that the multi-scenario
soybean growth dataset developed in this study effectively captured various production scenarios of soybeans, encompassing
both low and high yield simulations. The simulations of soybean yield followed a normal distribution. The mean value of
simulated soybean yield is 2675.66 kg ha$^{-1}$. The box plot illustrates the distribution of simulated data alongside statistics from
1980 to 2022, published yield data from literatures, and field measurements in 2022 and 2023. Compared with other yield data,
the simulated dataset for this study had the widest range of values which reflected the credibility of the multi-scenario in the
knowledge base and the effectiveness of the simulation results.



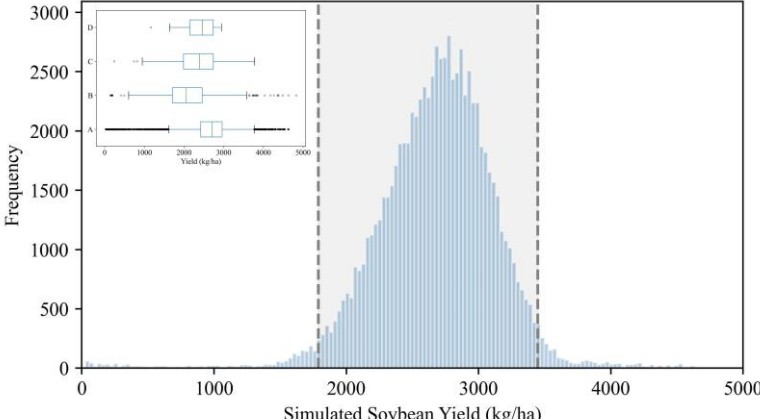

**Figure 5: Distribution of simulated soybean yield compared with other datasets. The gray area in the histogram represents 95 % confidence intervals. A represents simulated yield in this study (n = 83,972), B represents statistical yield from 1980 to 2022 (n = 961), C represents specific measurements from the literature (Chen et al., 2011; Fan et al., 2012; Liu et al., 2005, 2008; Liu and Herbert, 2002; Wang et al., 2020, 2024; Zheng and Zhang, 2021) (n = 138) and D represents measurements in 2022 and 2023 carried by this study (n = 39).**

## 4.2 Yield estimation at field scale

The performance of NortheastChinaSoybeanYield20m at field scale was validated using field measurement from 2022 and 2023 (Fig. 6). The comparison between estimated and observed yield showed a great equality with $R^2 > 0.65$ ($p < 0.01$) in both of the two years. The overall accuracy in estimations for both of the two years was 0.73 in $R^2$ ($p < 0.01$), 287.44 kg ha⁻¹ in RMSE and 10.02 % in MRE (Fig. A2), and the model achieved higher yield estimation accuracy in 2023 with RMSE of 271.07 kg ha⁻¹ and MRE of 8.57 % (Fig. 6b). This assessment indicated that the dataset well captured the spatial variation of soybean yield.

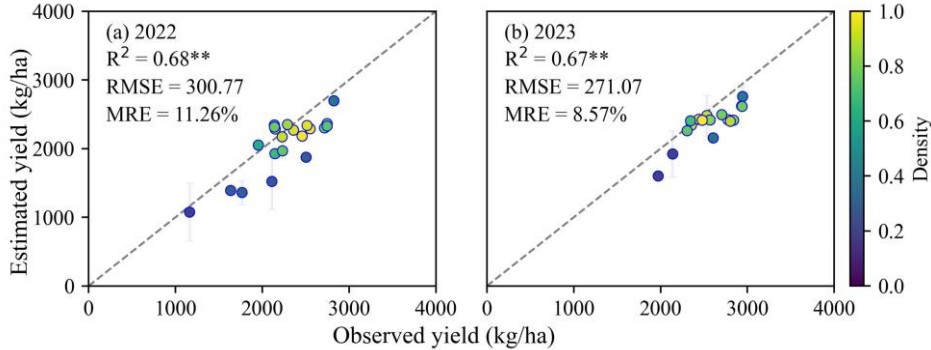

**Figure 6: Scatterplots between estimated and observed soybean yield in 2022 and 2023, respectively. The error-bars represent one standard deviation indicating the uncertainty of yield estimations. Dashed line represents 1:1 line. ** denotes statistical significance at p < 0.01.**

### 4.3 Yield estimation at regional scale

#### 4.3.1 Variability of accuracy through years

The accuracy assessment of NortheastChinaSoybeanYield20m at regional scale from 2019 to 2022 was presented in Fig. 7. The yield maps were aggregated to the municipal scale for comparison with statistics. It could be observed that the dataset achieved stable performance in different years. The correlation between annual estimated soybean yields and statistical yields were all above 0.60 (p < 0.01). The RMSE ranged from 221.69 kg ha$^{-1}$ to 310.66 kg ha$^{-1}$ and MRE from 8.24 % to 14.40 %. The result in 2022 yielded the highest accuracy with simulation error lower than 10 % (Fig. 7d). The overall accuracy in

estimations through the four years was 0.62 in $R^2$ (p < 0.01), 272.36 kg ha$^{-1}$ in RMSE and 12.08 % in MRE (Fig. 12a).

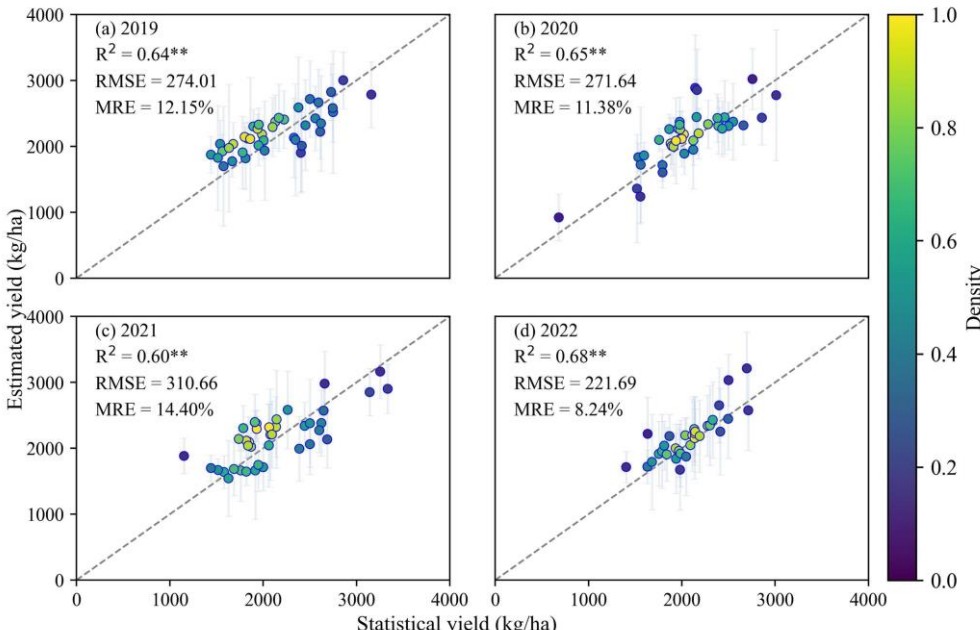

**Figure 7: Scatterplots between estimated soybean yield from Sentinel-2 and municipal statistical yields for 2019 – 2022 (excluding 2023 for which no statistical data was not available from the government). The error-bars represent one standard deviation indicating the uncertainty of yield estimations. Dashed line represents 1:1 line. ** denotes statistical significance at p < 0.01.**

For temporal performance of the NortheastChinaSoybeanYield20m at regional scale, the RMSE between model estimates and statistical yields in 2019-2022 was all lower than 500 kg ha$^{-1}$, with 80 % of cities having RMSE below 350 kg ha$^{-1}$ (Fig. 8a). Within the whole study area, the estimation error of soybean yield was larger in the northern part of Northeast China especially for the Greater Khingan Mountains area. Estimates in the relatively flat region, central of Northeast, showed less error. The spatial distribution pattern of MRE was similar to that of RMSE (Fig. 8b). The MRE was on average of 11.46 % for

all cities through the four years, indicating the great performance for the model to capture the interannual variability of soybean yield.



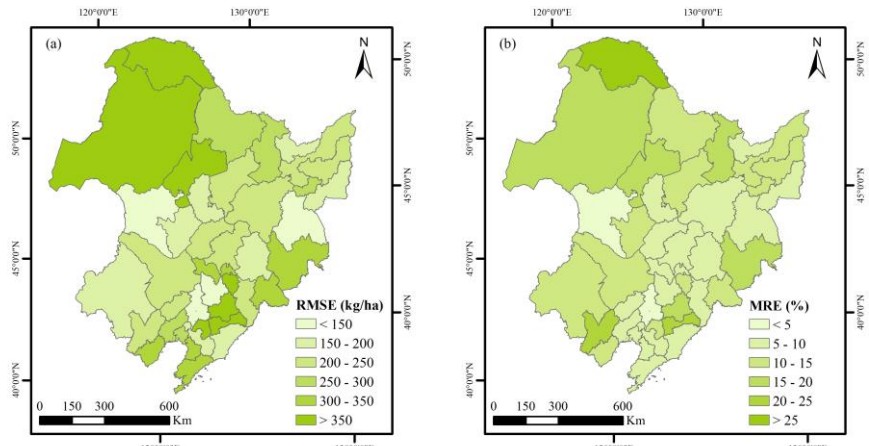

**Figure 8: Spatial patterns of the mean value of the root mean squared error and mean relative error between model estimated yields from Sentinel-2 and statistical yields from 2019 to 2022 (excluding 2023 for which no statistical data was not available from the government), (a) and (b), respectively. For years from 2019 to 2021, a total of 40 cities were calculated. For 2022, 32 cities were calculated due to missing statistics.**

### 4.3.2 Spatial-temporal dynamics of soybean yield

To analyze the spatial patterns of soybean yield in Northeast China, the distribution maps of soybean yield during 2019-2023 were generated (Fig. 9a-e). After calibration with MODIS LAI products, the Sentinel-2 seaming effect has been alleviated (Fig. 10) compared with the estimates before calibration (Fig. A3). The results showed that the estimation results had good spatial continuity. The soybean yield in Northeast China was mainly concentrated between 1500 and 2500 kg ha⁻¹. The soybean yield was generally higher in the central part of Northeast China where the terrain was relatively flat. The predicted yield is consistent with the statistical values (Table 5). Spatial variability could be observed (Fig. 9), characterized by the coefficient of variation (CV) ranging from 17.51 % to 29.65 % (Table 5).

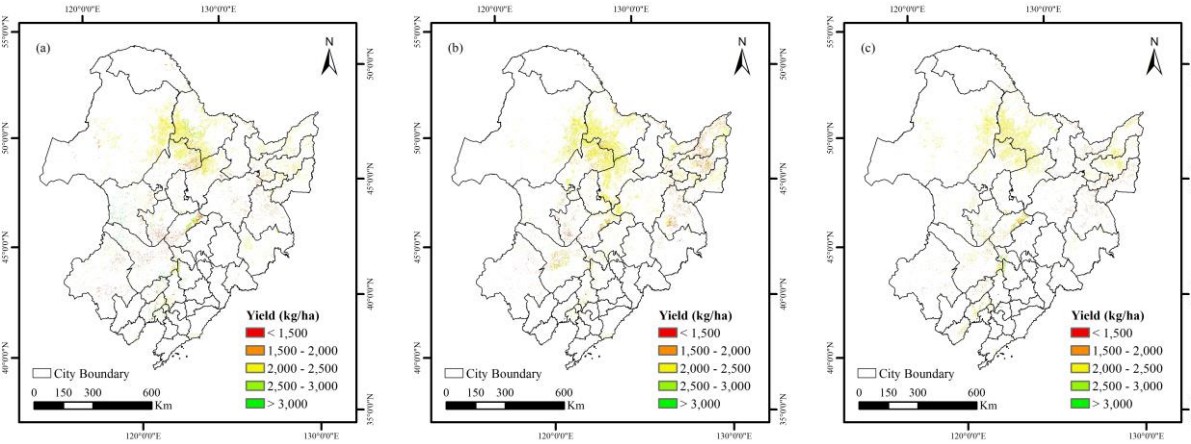





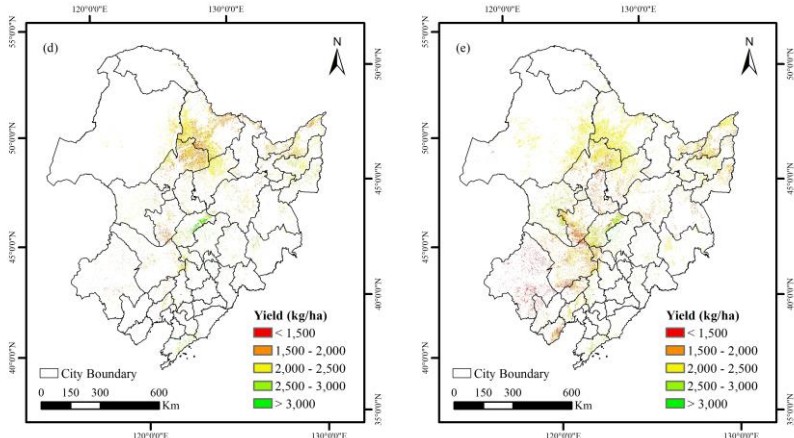

**Figure 9: Spatial distribution of annual soybean yield derived from Sentinel-2 after calibration in Northeast China from 2019 to 2023.**

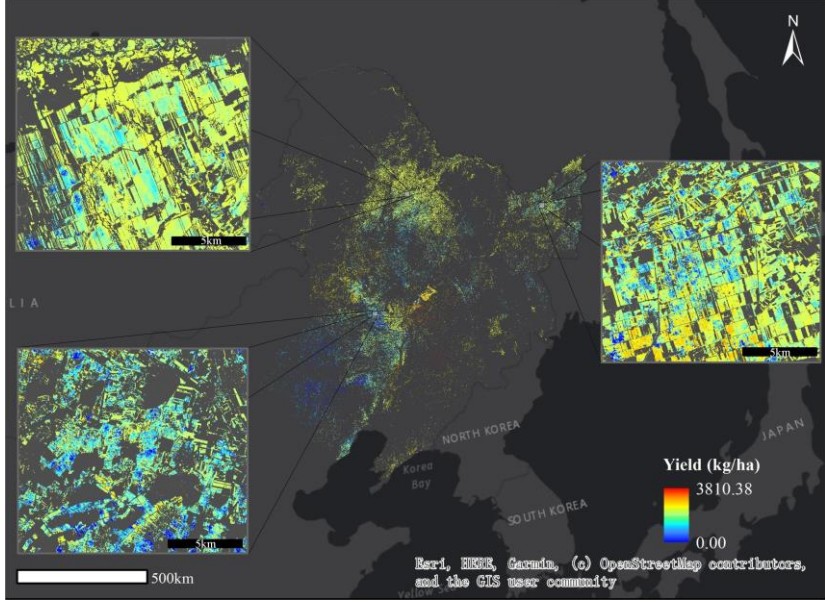

**Figure 10: An example of yield estimation result (for the year 2023) used to showcase detailed local estimates.**


**Table 5 Mean values of statistical soybean yield at municipal scale in Northeast China compared with mean values, standard deviation (STD) and coefficient of variation (CV) for estimated soybean yield in Northeast China.**

| Year | Statistics (kg ha⁻¹) | Mean (kg ha⁻¹) | STD (kg ha⁻¹) | CV (%) |
|------|------|------|------|------|
| 2019 | 2137.24 | 2150.02 | 504.61 | 23.47 |
| 2020 | 2069.08 | 2125.49 | 372.21 | 17.51 |
| 2021 | 2115.57 | 2136.65 | 374.58 | 17.53 |
| 2022 | 2073.68 | 2036.89 | 465.26 | 22.84 |
| 2023 | —— | 2035.34 | 603.43 | 29.65 |

For further study, we analyzed the spatial-temporal variation of soybean yield at the provincial scale (Fig. 11). The accuracy of soybean yield estimation at provincial scale was on average of 7.94 % in MRE (Fig. 11b). The accuracy of yield estimation at the provincial scale achieved the highest in 2022 (Fig. 11b), which was consistent with the results at the municipal scale (Fig. 7d). The soybean yield was highest in Liaoning Province through the four years. Although Heilongjiang Province had the largest soybean planting area in Northeast China, its soybean yield was at its lowest among different years (Fig. 11a). This could be attributed to the climate in Heilongjiang Province. The low temperatures may make it difficult to accumulate the required heat for normal development of soybean. Results showed that the soybean yield in the four provinces remained stable between 2019-2022, while in Jilin Province showed the most noticeable fluctuation, exhibiting a trend of initially decreasing and then increasing. The predicted yield further demonstrated the effectiveness of the method proposed in this study for capturing spatial-temporal variations in soybean production.

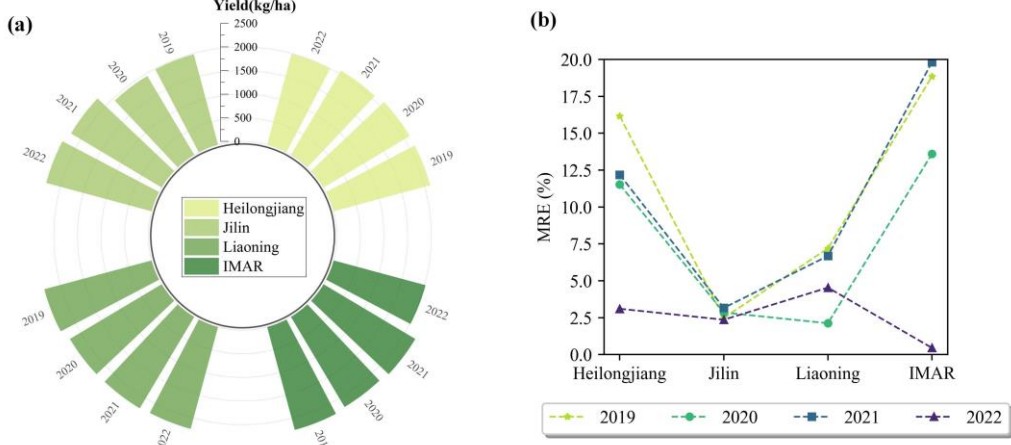

**Figure 11: Accuracy of the soybean yield estimation at provincial scale in Northeast China from 2019 to 2022. (a) represents the change in estimated yield for each province through the years; (b) represents MRE of results compared with statistical yield for each province**



## 5 Discussion

### 5.1 The complementarity between MODIS and Sentinel-2

In this study, both MODIS and Sentinel-2 satellite data were used forgenerating the soybean yield dataset. For further comparison, accuracy assessment was done for the estimations based on MODIS LAI at municipal scale. The overall accuracy in estimations for the four years was 0.58 in $R^2$ (p < 0.01), 272.36 kg ha$^{-1}$ in RMSE and 12.08 % in MRE (Fig. 12b), which was slightly lower than that using Sentinel-2. The uncertainty of yield estimation at municipal scale using MODIS data was higher than that using Sentinel-2. This might be due to that Sentinel-2 has a higher spatial resolution. In addition, in this study,

the seaming effect of Sentinel-2 was corrected using MODIS LAI products. The seaming effect of Sentinel-2 estimates had been greatly minimized (Fig. 10). Since calibration was done on a tile-by-tile basis rather than at each pixel, the original estimation details of yield maps were well preserved (Fig. 14). The results showed that the two data sources had the same ability to capture the spatial-temporal variation in soybean yield, yielding stable and similar prediction results across different years (Fig. 13). The estimation results in different years showed good correlation with statistical data, and the overall estimation

errors were less than 13 %.

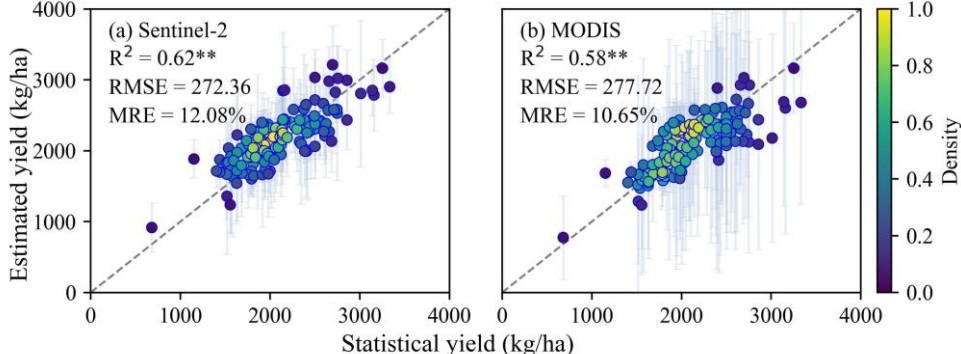

**Figure 12: Comparison between estimated and statistical yield for 2019 – 2022 using Sentinel-2 (a) and MODIS (b), respectively (excluding 2023 for which no statistical data was reported). The error-bars represent one standard deviation indicating the uncertainty of yield estimations. Dashed line represents 1:1 line. ** denotes statistical significance at p < 0.01.**

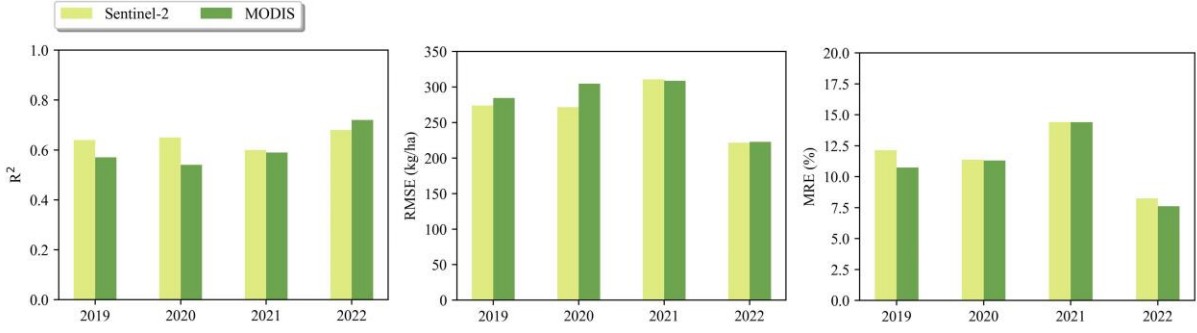


**Figure 13: Comparison of accuracy evaluation results for soybean yield estimation in 2019 – 2022 (excluding 2023, for which no statistical data was reported) using Sentinel-2 and MODIS data, respectively.**

In practical applications, better temporal resolution and spatial resolution are equally important to obtain ideal prediction results (Azzari et al., 2017). Since MODIS has a higher temporal resolution, it provides more available images during the crop

growing season to calculate the model input features ($LAI_{mean1}$ and $LAI_{mean2}$) more accurately. In addition, coarser spatial resolution can speed up spatial processing. However, our results showed that Sentinel-2 data with higher spatial resolution could be better to capture the spatial heterogeneity of yield among fields (Fig. 14). As the aim of this study was to generate yield dataset with high spatial resolution, MODIS LAI was only used to adjust the seaming effects of LAI derived from Sentinel-2 satellite. However, further integration of the information from remote sensing data with high spatial and temporal

resolution may form a more ideal data source for crop yield estimation (Gao and Anderson, 2019; He et al., 2018).

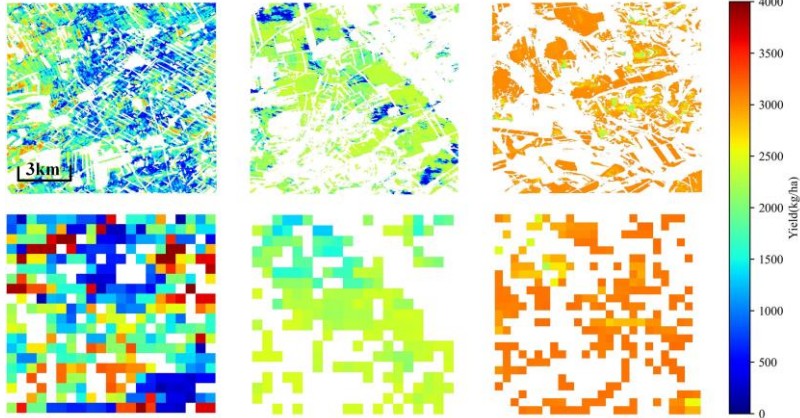

**Figure 14: Examples of soybean yield estimation at 3 km scale using Sentinel-2 (20 m) and MODIS (500 m) data, respectively.**

## 5.2 Limitations and future developments

Accurate monitoring of soybean yield is crucial for food policy decision-making and security assessment. Previous studies have

primarily focused on the impact of various factors (e.g., climate) on soybean yield (Guo et al., 2022; Zhao et al., 2023a). To our knowledge, high-resolution soybean yield dataset is currently unavailable in the main production regions of China. The study combined crop growth model with deep learning to construct a hybrid model driven by data and knowledge simultaneously for soybean yield estimation. The model retained its data mining capabilities while incorporating mechanistic constraints, thereby enhancing the model's interpretability and transferability. Accuracy verification based on in-situ and

statistical data showed that the NortheastChinaSoybeanYield20m generated in this study accurately estimated soybean yield at both field and regional scales (Fig. 6 and 7). However, there is still several work worth of advancement to further improve the accuracy in yield estimations.

a. Improvements in construction of soybean growth dataset





In this study, a multi-scenario soybean growth dataset was constructed by setting various simulation scenarios for the
input parameters of the WOFOST to provide training data for the yield estimation modeling. The model can easily be expanded
to regions and countries lacking for ground observation data (such as Africa and India) to provide agricultural monitoring in
the future. However, due to the extensive information in the simulation dataset, precise yield estimation came at the cost of
high computational demand. In addition, as the setting of model parameters mainly came from literatures in this study, the
construction of dataset might not fully account for all scenarios related to crop yield, such as pests, diseases and abiotic stresses
(Gaso et al., 2024).

In future research, it will be beneficial to improve the construction of soybean growth dataset to remove redundant
information and to better simulate various growing scenarios. For example, all of the meteorological data collected in this
study was used for simulation, which brought a lot of redundant information to the soybean growth dataset. In future study,
spatiotemporal clustering of meteorological station data can be carried out to further remove stations with redundant data. As
the growth of soybeans exhibit spatial variability in the study area, it is recommended to construct different dataset for different
ecological zones if more ground observation data is available in the future. The method of establishing ecological zones
includes establishing Tyson polygons based on meteorological stations (Huang et al., 2023).

b. Improvements in model estimations

Accurate estimation of soybean yield depended on the quality of the input data. The spatial resolution of remote sensing
imagery could limit the model's ability to predict spatial variability in yield. In this study, ERA5-land dataset was applied to
obtain the spatial-temporal distribution of soybean phenology in the study area. To be consistent with Sentinel-2 data, all
datasets were resampled to a 20 m resolution. Downscaling the coarse spatial data could increase the uncertainty of inputs to
the model. In addition, the presence of mixed pixels led to the loss of part of the surface information (Zhao et al., 2023) and
impaired the prediction ability of the model especially in the complex environment (Fig. 8). With the advent of higher spatial-
temporal resolution remote sensing data, the estimation accuracy of crop yield is expected to be further improved.

The mean values of LAI at two growth stages were used as model input features for regional soybean yield estimation.
The errors in LAI inversion from remote sensing data were introduced into the model prediction, thus increasing the uncertainty
of yield estimation. The coupling of agronomic knowledge and remote sensing mechanism has become a new research focus
(Chen et al., 2022; Hu et al., 2024). By coupling radiative transfer model such as PROSAIL (Jacquemoud et al., 2009)) with
crop growth model, remote sensing data can further provide additional constraints and enhance the simulation ability of crop
growth models (Ntakos et al., 2024). For example, the time series of crop biophysical variables simulated by crop growth
models can serve as input for radiative transfer models to generate corresponding spectral reflectance. Subsequently, these
spectral reflectance data can further be used as input features for deep learning models to estimate crop yield.

**6 Data availability**

The soybean yield dataset for Northeast China (NortheastChinaSoybeanYield20m) during the 2019-2023 period is available at https://doi.org/10.5281/zenodo.14263103 (Xu et al., 2024).

**7 Conclusions**

This study generated a 20 m soybean yield dataset in Northeast Chinda from 2019 to 2023 called NortheastChinaSoybeanYield20m using a hybrid method coupling crop growth model (WOFOST) with deep learning

algorithm (GRU). The construction of the hybrid method was based on a huge soybean growth dataset simulated by WOFOST accounting for various climates, crop varieties, soil types and agro-managements. The method effectively reduces the dependence on ground observation data, and had better spatial-temporal generalization.

The soybean yield dataset was generated using multi-source remote sensing data over the years 2019-2023. The results showed that the performance of the NortheastChinaSoybeanYield20m at field and regional scales was encouraging. The yield

estimations were highly consistent with in-situ measured and municipal statistical data ($p < 0.01$). The overall accuracy was 287.44 kg ha$^{-1}$ and 272.36 kg ha$^{-1}$ in RMSE, respectively. It was worth emphasizing that after correction with yield maps derived from MODIS LAI products, the seaming effect of Sentinel-2 was mitigated. The combined use of multi-source remote sensing data realized the complementarity of temporal resolution and spatial resolution. The results of estimation showed great spatial continuity. The dataset had stable yield estimation performance through different years, and could effectively capture

the spatial-temporal variation of soybean yield (MRE on average of 11.46 % for municipal scale and 7.94 % for provincial scale). Our proposed soybean yield dataset is helpful for optimizing soybean production distribution and ensuring food security.

**Authorship contributions**

JX (first author) and QL – conceptualization; JX (first author), XD, YZ, HW, JX, YS and YD – data curation; JX (first author), XD, TD – methodology; JX (first author), XD, JX and JZ – investigation; TD and QL – supervision; HW, JX (first author)

and JZ – validation; YZ, HW and JZ – visualization; JX (first author) – original draft preparation; XD, TD and YZ – reviewing and editing the manuscript.

**Competing interests**

The authors declare that they have no known competing financial interests or personal relationships that could have appeared to influence the work reported in this paper.



**Acknowledgements**

This research was funded by the National Key R&D Program of China (2021YFD1500103), the Strategic Priority Research Program of the Chinese Academy of Sciences (XDA28070504), the National Science Foundation of China (42371359), and the Key Program of High-resolution Earth Observation System (71-Y50G10-9001-22/23).

**Appendix A**

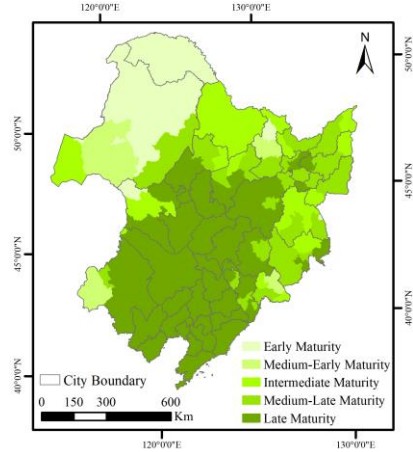


**Figure A1: Spatial distribution of soybean types in Northeast China.**

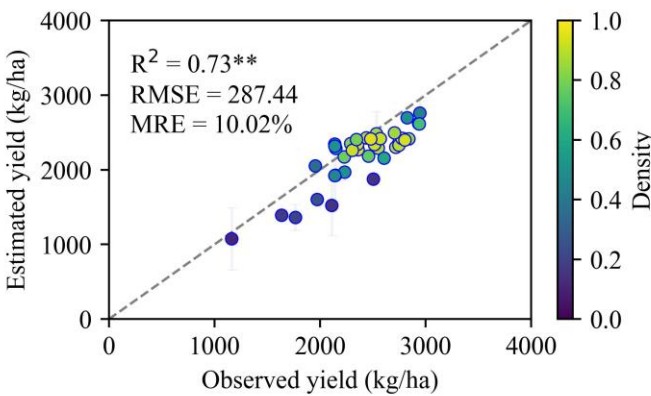

**Figure A2: Comparison between estimated and observed yield for both of 2022 and 2023. The error-bars represent one standard deviation indicating the uncertainty of yield estimations. Dashed lines represent 1:1 line. ** denotes statistical significance at p <**
**0.01.**




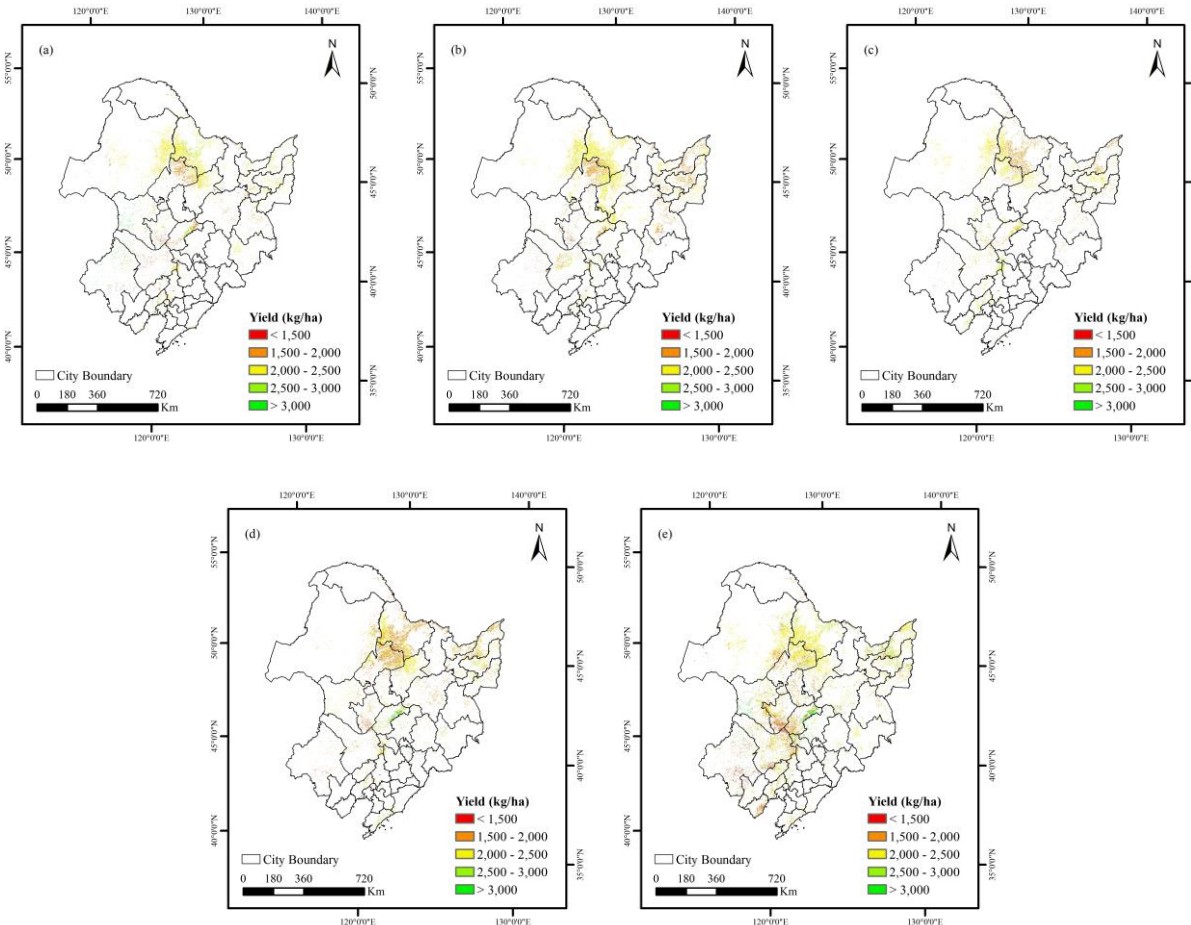

**Figure A3: Spatial distribution of annual soybean yield derived from Sentinel-2 before calibration in Northeast China from 2019 to 2023.**

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
