# Peer review of "NortheastChinaSoybeanYield20m: an annual soybean yield dataset at 20 m in Northeast China from 2019 to 2023"

_Earth System Science Data, 2024_

## Author Comment (AC1)

**Responses to the comments of Referee #1**

**Article ID:** essd-2024-586
**Title:** NortheastChinaSoybeanYield20m: an annual soybean yield dataset at 20 m in Northeast China from 2019 to 2023
**Authors:** Jingyuan Xu, Xin Du, Taifeng Dong, Qiangzi Li, Yuan Zhang, Hongyan Wang, Jing Xiao, Jiashu Zhang, Yunqi Shen, Yong Dong

Dear Reviewer,

Thank you very much for your thorough review and valuable comments on our manuscript. Your insightful feedback has significantly contributed to improving the quality and clarity of our work. In response to your suggestions, we have rigorously revised the manuscript to address each comment and suggestion. The key modifications include:

(1) Data Description Revisions: the data sections (e.g., field measurements, remote sensing data, and statistical sources) have been thoroughly revised to eliminate ambiguity and ensure reproducibility. Clarifications have been added to prevent misinterpretations on the datasets, including explicit dentitions of sampling protocols and spatial-temporal resolution.
(2) Methodological Enhancements: Additional details on the hybrid approach, such as descriptions of scenario parameterization, temporal feature extraction and computational have been added to strengthen methodological transparency and rigor.
(3) Readability Improvements: Key paragraphs and sentences have been restructured for logical flow. Figures or tables have been refined to align with revised text, ensuring visual clarity and consistency with results.

The detailed point-to-point responses are as follows. Texts in black are the reviewer's comments; those in blue are our responses to the reviewer's comments; and those in red and italics are the revised texts appeared in the revised manuscript.

We will attach a clean version (Manuscript_Clean_Version.docx) as well as a tracking enabled version (Manuscript_Marked_Version.docx) with editing marks for your reference.

I am very familiar with the WOFOST model and the dataset used by the author. It is not a good simulation project, not only because the simulation accuracy did not meet industry standards, but also because the author withheld many critical details and settings of the WOFOST in the manuscript, which makes it difficult for me to assess the rationality and scientific validity of the simulation. *Earth System Science Data*, as the name suggests, focuses on the application of datasets, but the author's professionalism in describing and processing the dataset is not good. Moreover, the description of CRU is severely inadequate. After reading the entire manuscript, I still do not understand the role of the CRU used by the author in this study.

**Reply:** Thank you for your thorough review and valuable feedback.

We sincerely appreciate your expertise and the time invested in evaluating our work. Below, we address your concerns point by point:

**(1) "The simulation accuracy did not meet industry standards, and critical details/settings of WOFOST were withheld, making it difficult to assess scientific validity."**

We acknowledge your concern and regret any lack of clarity about the WOFOST model and its reparameterization in our original manuscript.

In the revised manuscript, key edits to improve transparency include:
  (a) Additional Details Added: detailed description on the soil data (Section 2.2.3) and statistical data (Section 2.2.6) was added. We also have provided more detailed information to explicitly outline the WOFOST parameterization, including soil properties (Line 217 – 225), crop variety parameters (e.g., TSUM1, TSUM2, Table A1), and agro-management scenarios (e.g., planting date, Line 238 – 245).
  (b) Model Calibration and Validation: we have revised the section 5.2 to include more detailed comparisons with existing studies at various spatial scales. For instance, our accuracy at the municipal-scale was RMSE of 272.36 kg ha$^{-1}$ outperformed the 420 kg ha$^{-1}$ reported by Von Bloh et al., (2023). Additionally, we have expanded the discussion on the comparison of our dataset with soybean yield datasets from other countries, demonstrating the higher precision of our estimates. The comparisons include a detailed analysis of RMSE, supporting the claim that our method provides reliable and accurate yield estimates (Line 469 – 478).
  (c) Supplementary Material: A table of WOFOST input parameters is provide in Supplementary Table S1 to enhance reproducibility.

**(2) "The description of CRU is severely inadequate, and the dataset's role in the study is unclear."**

We apologize for the initial oversight.

In the revised manuscript, we have enhanced the description on the WOFOST- GRU integration. We have detailed and clarified the coupling mechanism between the GRU and WOFOST models including generating input features from time-series LAI data, and explicitly outlining the training workflow (Section 3.2, Line 264 – 279). Figure 2 has been revised to explicitly illustrate the flow of data between WOFOST simulations, remote sensing inputs, and GRU-based yield estimation.

**(3) *"The dataset description lacks professionalism, and the manuscript does not align with Earth System Science Data's focus."***

We have rigorously revised the manuscript to emphasize: (a) Dataset Documentation, the "data availability" has been revised to detail file formats, data spatial-temporal resolution. Metadata with these information has been added to the dataset hosted on Zenodo. (b) Application Focus, we have revised the conclusions to synthetically illustrate the innovation of the method for dataset construction and demonstrate the dataset's utility for agricultural management practise (e.g., optimizing sowing date for maxing yield), aligning with the journal emphasis on actionable Earth system data.

We are grateful for your critique, which has significantly strengthened our work. We hope these revisions address your concerns raised and significantly enhance the quality and clarity of the manuscript.

Below, we provide a detailed point-by-point response to address the specific concerns you raised:

1. The study spanned from 2019 to 2023, but the sampling data was only from 2022 and 2023 (Fig.1). The author should explain this issue in the text.

**Reply:** Thank you very much for the suggestion.

In the revised manuscript, we have clarified that although the study spanned from 2019 to 2023, field observations were not conducted from 2019 to 2021 due to resource limitations. As a result, the sampling data for this study were collected only in 2022 and 2023 (Line 106 - 108).

*Due to resources and personnel constraints, in-situ yield measurements were unavailable for the initial years (2019 - 2021). Field-scale yield data was collected through field surveys in September 2022 and 2023, with 21 and 18 sample plots surveyed annually, respectively.*

2. The soil data should be described in more detail, for example, which soil parameters were used in this study.

**Reply:** Thanks for your suggestion.

As suggested, we have provided a more detailed description of the soil data used in this study. For our study, we did not use the soil attribute data (such as chemical characteristics) in this analysis, focusing only on the spatial distribution of soil types to characterize the soil environment in the study region. This clarification has been added to the revised manuscript (Section 2.2.3).

*Soil data was obtained from the 1:1000,000 Chinese Soil Database, accessible via the Geographic Data Sharing Infrastructure and the Global Resources Data Cloud platform (www.gis5g.com). The dataset adopted the traditional "Chinese soil classification system", with subcategories serving as the foundational mapping unit. It categorized soil hierarchically into 12 soil orders, 61 soil types, and 227 subcategories, providing comprehensive coverage of China's soil diversity. The dataset consisted of two components: spatial data for vector layers delineating soil type distributions at the national scale, and attribute data that include key chemical properties (e.g., pH, organic matter) and physical characteristics (e.g., texture, bulk density). In this study, the 1:1000,000 soil spatial data was obtained to map soil types for the study area, enabling integration with agro-ecological variables (e.g., crop suitability, irrigation requirements) in the hybrid yield estimation framework.*

3. The author used statistical data from 1980 to 2022, but the study's time scale is from 2019 to 2023. This is confusing for the readers. Please provide an explanation.

**Reply:** Thanks for your suggestion.

We have clarified the use of statistical data in the revision. In this study, the statistical data served two main purposes. Firstly, to ensure that the multi-scenarios soybean growth dataset, which was constructed in this study, accurately represented a wide range of soybean production conditions, the statistical data from 1980 to 2022 were all used to validate the reasonableness of the model simulations. Secondly, statistical data from 2019 to 2022 were specifically applied to evaluate accuracy of soybean yield estimation at the regional scale.

This distinction has been clearly explained in the revised manuscript to avoid any confusion for the readers (Section 2.2.6).

*The crop yield records (1980-2022) were obtained from the Statistical Yearbook published by provincial authorities: Heilongjiang (http://tjj.hlj.gov.cn), Jilin (http://tjj.jl.gov.cn), Liaoning (https://tjj.ln.gov.cn) , and Inner Mongolia Autonomous Region (IMAR , https://tj.nmg.gov.cn). Due to the incomplete publication of the 2022 Statistical Yearbook, municipal-level yield for that year were limited to a subset of cities. The statistical data were used for model simulation validation and regional-*

*scale yield accuracy evaluation. The multi-scenario soybean growth dataset, simulated using multi-years meteorological data, various soil types, multiple soybean varieties and different agro-managements, required validation against real production conditions. The historical statistics (1980 – 2022), supplemented by published yield datasets and field samples, were used to evaluate the plausibility of the simulated dataset validated the reasonableness of the model simulations. For validating yield estimates at regional scale, data from 2019 to 2022 employed to quantify estimation errors at regional scales.*

4. The technology roadmap that needs improvement. 1) The author mentioned agro-management data in Figure 2, but it is not mentioned in Section 2.2 Data collections. 2) The sampling data mentioned in the Data collections section is not reflected in the figures, as well as meteorological data from National Meteorological Information Center. 3) The method of combining remote sensing data and model output through GRU is described too simplistically. 4) The author allocates a large proportion of the figures to how WOFOST conducts simulations, but this is not the focus of this study. The focus of this study should be on how to use models and remote sensing coupled for yield estimation, just as the author introduces in the research objective: "Designing a hybrid model coupling crop growth model and deep learning model for soybean yield estimation." The technology roadmap should more detailed display the research focus.

**Reply:** Thank you for pointing this out.

We have made substantial revisions to improve the technology roadmap, addressing the points raised:
(1) The study primarily simulated different soybean agro-management scenarios by setting different planting dates as literatures have highlight the importance of planting date on soybean yield. In addition, for large-scale estimation modeling, it's more practical to focus on broader factors like planting dates rather than detailed management measures. The agro-management data was collected alongside in-situ measurements. We have now included an explanation of the agro-management data collection process in Section 2.2.1 (Line 112 - 114) to ensure consistency between the figure and the text.

*Due to resources and personnel constraints, in-situ yield measurements were unavailable for the initial years (2019 - 2021). Field-scale yield data was collected through field surveys in September 2022 and 2023, with 21 and 18 sample plots surveyed annually, respectively (Fig. 1). Each plot covered an area of approximately 100 m × 100 m, within which nine randomly distributed 1 m × 1 m quadrats were selected for destructive soybean yield sampling. The central geo-location of each quadrat was recorded using a GPS device with accuracy of 1 m. Harvested beans for each quadrat were then oven-dried 72 hours to determine moisture-free yield, which was processed at the Hailun Agricultural Ecology Experimental Station, Chinese*

*Academy of Sciences. Finally, the mean yield of the nine quadrats was calculated to represent the plot-level yield.* ***Additionally, soybean planting dates across regions were collected through field surveys to support agro-management parameterization in the model.***

(2) The sampling data were used to validate soybean yield estimation accuracy at the field scale, while the meteorological data from the National Meteorological Information Center were used as input for the WOFOST model to provide essential weather parameters. Both of these datasets are represented in the revised figure (field measured samples and meteorological station data, respectively), and their roles have been explicitly described in the text.

(3) We have revised the flowchart to better represent the hybrid model's structure. The updated version now clearly distinguishes two main components: the first part describes the construction of the hybrid model, combining WOFOST and GRU for yield estimation, while the second part focuses on how remote sensing data are used in conjunction with this hybrid model to estimate soybean yield at the regional scale. We have provided more detail in the flowchart regarding how remote sensing data are integrated as inputs into the GRU model and how they contribute to spatial yield predictions.

(4) We agree that the focus of the study should be on the coupling of remote sensing data with crop growth models for yield estimation, as stated in the research objectives. Therefore, the updated technology roadmap places greater emphasis on this aspect, reducing the proportion dedicated to the WOFOST model and highlighting how the hybrid model (WOFOST + GRU) is used for yield estimation. This revision better aligns the roadmap with the research focus and research objectives, as well as the methodology described in the manuscript.

[Figure]

**Figure 2: The flowchart of the overall yield estimation methodology in this study.**

5.  Which sub-model of PCSE did the author use? LINTUL3 or Wofost72_PP?

**Reply:** Thanks for your suggestion.

Our soybean simulations were conducted using the Wofost72_WLP_CWB (water-limited mode) within the PCSE (v5.5.) for soil-water dynamics and crop responses to water stress. This is because rainfed agricultural practices dominant in Northeast China, where water availability rather than nutrient constraints is the major limiting factor on crop growth.
Relevant edits have been revised in the revision (Line 197 – 199).

*In this study, the World Food Studies Simulation Model (WOFOST) model (Diepen et al., 1989) was employed to generate knowledge-based soybean growth dataset.*

*Detailed information about the model could be found in Huang et al., (2015). **Since rainfed agricultural practices dominant in Northeast China, the water-limited mode of the WOFOST (Wofost72_WLP_CWB) was employed for soybean simulation from the Python Crop Simulation Environment (PCSE) framework version 5.5.***

6. As far as I know, VAP is not included in the ERA5 dataset. How did the author obtain the VAP data?

**Reply:** Thank you for pointing out this.

The study only used the daily aggregated air temperature data at 2 meters from the ERA5 dataset to calculate the spatial distribution of soybean phenology, which was used to guide the acquisition time for remote sensing data. For the meteorological parameters required by the WOFOST model (including VAP), all data were sourced from meteorological station data rather than ERA5.

This clarification has been added to the manuscript in Line 132 – 137 and Line 211 – 212.

***The climate reanalysis data was obtained from the ERA5-land Daily Aggregated ECMWF product. The data was only used to calculate soybean phenology for preparation of yield estimations.** It was a global climate reanalysis product that provided continuous climate data at a resolution of 0.1° × 0.1° (e.g., air temperature and atmospheric pressure) starting from 1950. The daily aggregated air temperature data at 2 m above the surface of land measured in kelvin (K) **during the soybean growth periods from 2019 to 2023 was collected in this study** from the Google Earth Engine (http://earthengine.google.com). The product was then resampled to 20 m using bilinear interpolation model.*

*The meteorological parameters required in WOFOST is shown in Table 1. **The data was only provided by meteorological stations.** To account for various meteorological conditions, the meteorological data collected from the selected 51 meteorological stations over a period of 42 years (1980-2021) was all utilized to provide values of these parameters. The meteorological inputs were then processed into the format recognized by the model.*

7. Line 209-215: The description of the calculation process for soil parameters is too simplistic; a detailed calculation process should be provided. For example, which parameters from the Chinese soil database were used in the study, and what theories/formulas were utilized to calculate the SMW, SMFCF, SM0, and K0 required by the WOFOST model? Is Table 2 a lookup table? Where did it come from?

**Reply:** Thanks for your suggestion.

In the revised version of the manuscript, we have provided more detailed information on the soil parameter settings (Line 217 – 225). Based on the soil spatial data, we found that the main soil types in the study area can be categorized as sandy loam, light loam, medium loam, and heavy loam. The parameter settings for these different soil types were primarily obtained from existing literature. Table 2 is not a lookup table, but a compilation of the parameters based on previous studies.

We have now indicated the sources of the soil parameters presented in Table 2 in the manuscript.

*The soil parameters in the WOFOST mainly include soil moisture content at wilting point (SMW), field capacity (SMFCF) and saturation (SM0) as well as hydraulic conductivity of saturated soil (K0). According to the 1:1000,000 Chinese soil database, the soil texture in study area is predominantly loam soil, which can be further divided into sandy loam, light loam, medium loam and heavy loam. Sun et al., (2022) found that the properties of typical heavy loam suitable for soybean cultivation were similar to those provided in the model source file WOFOST Control Centre/SOILD/EC2.NEW and further updated the soil parameters of heavy loam based on this file by incorporating soil data observed at meteorological stations. The parameters of sandy loam, light loam and heavy loam were collected from Du et al., (2025). The value settings of soil parameters for different soil types in the study area was presented in Table 2 (Du et al., 2025; Sun et al., 2022). They were all used as model inputs to consider the impact of different soil types on soybean production.*

8. Line 215: The description of Table 3 is redundant. It suffices to directly list the values and sources of the WOFOST crop parameters. Table 4 should list all crop parameters in WOFOST, not just the main crop parameters.

**Reply:** Thanks for your suggestion. We have removed Table 3 and its accompanying description. Instead, we have provided a complete list of the WOFOST crop parameters and their corresponding values as suggested. Due to the length of the table, it has been placed in the appendix (Table A1). This change ensures a more comprehensive and concise presentation of the crop parameters used in the study.

*In this study, the soybeans were classified into five types including early maturity, medium-early maturity, intermediate maturity, medium-late maturity and late maturity according to Qu et al., (2023). In the WOFOST model, soybean growth stages are mainly determined by temperature-related parameters including the minimum and maximum threshold temperature for emergence (TBASEM, TEFFMX, respectively), accumulated temperature (Te) from sowing to emergence (TSUMEM), from emergence to anthesis (TSUM1) and from anthesis to maturity (TSUM2). The accumulated temperature for different growth stage is sensitive to crop varieties. They were set according to the historical meteorological data and observation data of*

*soybean and had been validated using actual measurements of soybean development periods in Qu et al., (2023). Other crop parameters were set used the default values of the WOFOST model or the optimal values from the study of Sun et al., (2022). Values of all crop parameters could be found in Table A1.*

**Table A1 Values of crop parameters in WOFOST.**

| Parameter | Description | Units | Value | Source |
|---|---|---|---|---|
| Crop initial parameters | | | | |
| TDWI | Initial total crop dry weight | kg ha$^{-1}$ | 120 | Default value in WOFOST |
| RGRLAI | Maximum relative increase in LAI | ha ha$^{-1}$ d$^{-1}$ | 0.01 | Default value in WOFOST |
| Parameters for emergence | | | | |
| TBASEM | Minimum threshold temperature for emergence | ℃ | 8.0 | Qu et al., (2023) |
| TEFFMX | Maximum threshold temperature for emergence | ℃ | 22.0 | Qu et al., (2023) |
| TSUMEM | Accumulated temperature from sowing to emergence | ℃ | 70.0 | Qu et al., (2023) |
| Phenological parameters | | | | |
| DLO | Optimal daylength for development | h | -99 | Default value in WOFOST |
| DLC | Critical daylength | h | -99 | Default value in WOFOST |
| TSUM1 | Cumulative temperature from emergence to anthesis | ℃ | 450 (early maturity) 480 (medium-early maturity) 520 (intermediate maturity) 540 (medium-late maturity) 580 (late maturity) | Qu et al., (2023) |
| TSUM2 | Cumulative temperature from anthesis to maturity | ℃ | 660 (early maturity) 770 (medium-early maturity) 870 (intermediate maturity) 960 (medium-late maturity) 1000 (late maturity) | Qu et al., (2023) |
| Green area parameters | | | | |
| TBASE | Lower threshold temperature for aging of leaves | ℃ | 7.0 | Default value in WOFOST |
| SPAN | Life span of leaves growing at 35 ℃ | d | 23 | Default value in WOFOST |
| SLATB00 | Specific leaf area at DVS = 0.00 | ha kg$^{-1}$ | 0.00140 | Default value in WOFOST |
| SLATB045 | Specific leaf area at DVS = 0.45 | ha kg$^{-1}$ | 0.00250 | Default value in WOFOST |
| SLATB090 | Specific leaf area at DVS = 0.90 | ha kg$^{-1}$ | 0.00250 | Default value in WOFOST |
| SLATB200 | Specific leaf area at DVS = 2.00 | ha kg$^{-1}$ | 0.00070 | Default value in WOFOST |
| Assimilation parameters | | | | |

| | | | | |
|---|---|---|---|---|
| KDIFTB00 | Extinction coefficient for diffuse visible light (DVS = 0) | - | 0.80 | Default value in WOFOST |
| KDIFTB200 | Extinction coefficient for diffuse visible light (DVS = 2) | - | 0.80 | Default value in WOFOST |
| EFFTB0 | Light use efficiency of a single leaf (T = 0 °C) | $kg\ ha^{-1}\ h^{-1}\ J^{-1}\ m^2\ s^{-1}$ | 0.40 | Default value in WOFOST |
| EFTB40 | Light use efficiency of a single leaf (T = 40 °C) | $kg\ ha^{-1}\ h^{-1}\ J^{-1}\ m^2\ s^{-1}$ | 0.40 | Default value in WOFOST |
| AMAXTB00 | Maximum leaf $CO_2$ assimilation rate (DVS = 0) | $kg\ ha^{-1}\ h^{-1}$ | 29.00 | Default value in WOFOST |
| AMAXTB170 | Maximum leaf $CO_2$ assimilation rate (DVS = 1.7) | $kg\ ha^{-1}\ h^{-1}$ | 25.31 | Sun et al., (2022) |
| AMAXTB200 | Maximum leaf $CO_2$ assimilation rate (DVS = 2) | $kg\ ha^{-1}\ h^{-1}$ | 0.00 | Default value in WOFOST |
| TMPFTB00 | Reduction factor of AMAX (T = 0 °C) | - | 0.00 | Default value in WOFOST |
| TMPFTB10 | Reduction factor of AMAX (T = 10 °C) | - | 0.30 | Default value in WOFOST |
| TMPFTB20 | Reduction factor of AMAX (T = 20 °C) | - | 0.60 | Default value in WOFOST |
| TMPFTB25 | Reduction factor of AMAX (T = 25 °C) | - | 0.80 | Default value in WOFOST |
| TMPFTB30 | Reduction factor of AMAX (T = 30 °C) | - | 1.00 | Default value in WOFOST |
| TMPFTB35 | Reduction factor of AMAX (T = 35 °C) | - | 1.00 | Default value in WOFOST |
| Conversion of assimilates into biomass | | | | |
| CVL | Conversion efficiency of assimilates into leaf tissue | $kg\ kg^{-1}$ | 0.72 | Default value in WOFOST |
| CVO | Conversion efficiency of assimilates into storage organs | $kg\ kg^{-1}$ | 0.48 | Default value in WOFOST |
| CVR | Conversion efficiency of assimilates into root tissue | $kg\ kg^{-1}$ | 0.72 | Default value in WOFOST |
| CVS | Conversion efficiency of assimilates into stem tissue | $kg\ kg^{-1}$ | 0.69 | Default value in WOFOST |
| Maintenance respiration parameters | | | | |
| Q10 | Relative change in respiration rate per 10 °C temperature increase | - | 2.0 | Default value in WOFOST |
| RML | Ralative maintenance respiration rate of leaves | $kg\ CH_2O\ kg^{-1}\ d^{-1}$ | 0.03 | Default value in WOFOST |
| RMO | Ralative maintenance respiration rate of storage organs | $kg\ CH_2O\ kg^{-1}\ d^{-1}$ | 0.017 | Default value in WOFOST |
| RMR | Ralative maintenance respiration rate of toots | $kg\ CH_2O\ kg^{-1}\ d^{-1}$ | 0.01 | Default value in WOFOST |
| RMS | Ralative maintenance respiration rate of stems | $kg\ CH_2O\ kg^{-1}\ d^{-1}$ | 0.015 | Default value in WOFOST |
| Partitioning parameters | | | | |
| FRTB00 | Fraction of total dry matter to roots at DVS = 0 | $kg\ kg^{-1}$ | 0.62 | Sun et al., (2022) |
| FRTB075 | Fraction of total dry matter to roots at DVS = 0.75 | $kg\ kg^{-1}$ | 0.35 | Default value in WOFOST |
| FRTB100 | Fraction of total dry matter | $kg\ kg^{-1}$ | 0.15 | Default value in |

| | | | | |
|---|---|---|---|---|
| | to roots at DVS = 1 | | | WOFOST |
| FRTB150 | Fraction of total dry matter to roots at DVS = 1.5 | kg kg$^{-1}$ | 0.00 | Default value in WOFOST |
| FRTB200 | Fraction of total dry matter to roots at DVS = 2.0 | kg kg$^{-1}$ | 0.00 | Default value in WOFOST |
| FLTB00 | Fraction of total dry matter to leaves at DVS = 0 | kg kg$^{-1}$ | 0.70 | Default value in WOFOST |
| FLTB100 | Fraction of total dry matter to leaves at DVS = 1.0 | kg kg$^{-1}$ | 0.70 | Default value in WOFOST |
| FLTB115 | Fraction of total dry matter to leaves at DVS = 1.15 | kg kg$^{-1}$ | 0.60 | Default value in WOFOST |
| FLTB130 | Fraction of total dry matter to leaves at DVS = 1.3 | kg kg$^{-1}$ | 0.43 | Default value in WOFOST |
| FLTB150 | Fraction of total dry matter to leaves at DVS = 1.5 | kg kg$^{-1}$ | 0.15 | Default value in WOFOST |
| FLTB200 | Fraction of total dry matter to leaves at DVS = 2.0 | kg kg$^{-1}$ | 0.00 | Default value in WOFOST |
| FSTB00 | Fraction of total dry matter to stems at DVS = 0 | kg kg$^{-1}$ | 0.30 | Default value in WOFOST |
| FSTB100 | Fraction of total dry matter to stems at DVS = 1.0 | kg kg$^{-1}$ | 0.30 | Default value in WOFOST |
| FSTB115 | Fraction of total dry matter to stems at DVS = 1.15 | kg kg$^{-1}$ | 0.25 | Default value in WOFOST |
| FSTB130 | Fraction of total dry matter to stems at DVS = 1.3 | kg kg$^{-1}$ | 0.10 | Default value in WOFOST |
| FSTB150 | Fraction of total dry matter to stems at DVS = 1.5 | kg kg$^{-1}$ | 0.10 | Default value in WOFOST |
| FSTB200 | Fraction of total dry matter to stems at DVS = 2.0 | kg kg$^{-1}$ | 0.00 | Default value in WOFOST |
| FOTB00 | Fraction of total dry matter to storage organs at DVS = 0 | kg kg$^{-1}$ | 0.00 | Default value in WOFOST |
| FOTB100 | Fraction of total dry matter to storage organs at DVS = 1.0 | kg kg$^{-1}$ | 0.00 | Default value in WOFOST |
| FOTB115 | Fraction of total dry matter to storage organs at DVS = 1.15 | kg kg$^{-1}$ | 0.15 | Default value in WOFOST |
| FOTB130 | Fraction of total dry matter to storage organs at DVS = 1.3 | kg kg$^{-1}$ | 0.47 | Default value in WOFOST |
| FOTB150 | Fraction of total dry matter to storage organs at DVS = 1.5 | kg kg$^{-1}$ | 0.75 | Default value in WOFOST |
| FOTB200 | Fraction of total dry matter to storage organs at DVS = 2.0 | kg kg$^{-1}$ | 1.00 | Default value in WOFOST |
| **Death rate parameters** | | | | |
| PERDL | Maximum relative death rate of leaves due to water stress | kg kg$^{-1}$ d$^{-1}$ | 0.03 | Default value in WOFOST |
| RDRRTB00 | Relative death rate of roots at DVS = 0 | kg kg$^{-1}$ d$^{-1}$ | 0.00 | Default value in WOFOST |
| RDRRTB150 | Relative death rate of roots at DVS = 1.5 | kg kg$^{-1}$ d$^{-1}$ | 0.00 | Default value in WOFOST |
| RDRRTB151 | Relative death rate of roots at DVS = 1.51 | kg kg$^{-1}$ d$^{-1}$ | 0.02 | Default value in WOFOST |
| RDRRTB200 | Relative death rate of roots at DVS = 2.0 | kg kg$^{-1}$ d$^{-1}$ | 0.02 | Default value in WOFOST |
| RDRSTB00 | Relative death rate of stems | kg kg$^{-1}$ d$^{-1}$ | 0.00 | Default value in |

| | | | | | |
|---|---|---|---|---|---|
| | at DVS = 0 | | [1] | | WOFOST |
| RDRSTB150 | Relative death rate of stems at DVS = 1.5 | kg kg-1 d-[1] | | 0.00 | Default value in WOFOST |
| RDRSTB151 | Relative death rate of stems at DVS = 1.51 | kg kg-1 d-[1] | | 0.02 | Default value in WOFOST |
| RDRSTB200 | Relative death rate of stems at DVS = 2.0 | kg kg-1 d-[1] | | 0.02 | Default value in WOFOST |
| Water use parameters | | | | | |
| CFET | Correction factor transpiration rate | - | | 1.0 | Default value in WOFOST |
| DEPNR | Crop group number for soil water depletion | - | | 5.0 | Default value in WOFOST |
| IAIRDU | Air ducts in roots present (=1) or not (=0) | - | | 0 | Default value in WOFOST |
| IOX | Oxygen stress effect enabled (=1) or not (=0) | - | | 0 | Default value in WOFOST |
| Rooting parameters | | | | | |
| RDI | Initial rooting depth | cm | | 10 | Default value in WOFOST |
| RRI | Maximum daily increase in rooting depth | cm d-1 | | 1.2 | Default value in WOFOST |
| RDMCR | Maximum rooting depth | cm | | 120 | Default value in WOFOST |

9. Line 235: What's the setting of the fertilizer application rate and timing in the WOFOST?

**Reply:** Thanks for your suggestion.

We would like to clarify that in this study, no fertilizer applications were considered. For agricultural production management, literatures have highlight that the planting date has a significant impact on soybean yield. The study simulated different agricultural management practices by varying the planting dates. As soybeans in study area are seldom subjected to nutrient stress according to the field surveys, the WOFOST model was not configured with specific fertilizer application rates or timings for this study. As when it comes to large-scale estimation modeling, focusing too much on detailed management measures might not be practical. A more generalized approach, taking into account broader factors like planting dates, can provide a better balance between accuracy and feasibility for large-area predictions.

 This has been explained in the revised manuscript (Line 238 – 245).

*Research demonstrates that planting date significantly influences soybean yield components, including pods per plant, seed count, and plant biomass, while also affecting oil and protein content (Urda et al., 2024). When conducting large-scale estimation modelling, prioritizing overly detailed management practices may lack practicality. Furthermore, field surveys indicate that soybeans in study area rarely experience nutrient stress. Given these findings, the simulations did not account for fertilizer applications. To balance realism and scalability, agro-management variability was represented solely through planting date adjustments. Since soybeans*

*in the study area are typically sown between late April and late May, so simulations incorporated four planting scenarios: 20 April, 30 April, 10 May, and 20 May. Overall, the simulations capture phenological variability while avoiding unnecessary complexity in regional scale.*

10. Line 244: After reading Section 3.2 Development of the Grated Recurrent Unit model (GRU), I am still unclear about the role of GRU in this study. The author's explanation of the principles of GRU is unclear. It does not directly describe how GRU combines the output of the WOFOST model with remote sensing data, as shown in the technical roadmap. Figure 3 lacks self-explanatory power, leaving it unclear what exactly the inputs and outputs of the GRU are.

**Reply:** Thanks for your suggestion. In the revised version, we further clarified the role of the GRU model in this study. The GRU was used for large-scale soybean yield estimation in this study. Since the internal structure of the GRU model was not adjusted in this study, we reduced the description of the GRU model's principles and instead focused more on its application in yield estimation. We removed the description of the GRU cell structure in Figure 3 and replaced it with a more detailed explanation of how the GRU model was trained using the soybean growth dataset simulated by the WOFOST model, enabling it to quickly estimate soybean yield at the regional scale. Additionally, we specified the inputs and outputs of the GRU model. In this study, the GRU model uses the average LAI values at different soybean growth stages as input features and soybean yield as the output. The trained GRU model was then integrated with remote sensing data for large-scale soybean yield mapping using the features derived from the remote sensing data as inputs (Section 3.2).

*After construction of soybean growth dataset, the GRU model was used for large-scale soybean yield estimation. The GRU is a type of RNN model similar to LSTM. It controls the flow of information through two gates, update and reset gates (Cho et al., 2014). The update gate aims to control how much of the past information that are retrained and will be used in the future calculation. The reset gate aims to evaluate whether the remained previous information can be ignored in the new candidate hidden state. The use of two gates maintains the balance between retaining the hold hidden state and incorporating new information (Peng and Yili, 2022; Zhang et al., 2022). This improves the training speed of the model and helps mitigate the vanishing gradient problem during training. Since GRU can effectively capture long-term dependencies in time series data, it has achieved good performance in applications of crop yield estimation (Gopi and Karthikeyan, 2023; Ren et al., 2023b). The model was constructed based on TensorFlow 2.6 in this study. More details about GRU could be found in Cho et al., (2014).*

*The GRU model was trained based on the multi-scenario soybean growth dataset generated by the WOFOST model. We firstly determined the features sensitive to soybean yield, and then extracted the corresponding features and soybean yield from soybean growth dataset, which were used as the GRU's input*

*and output to train the GRU model. Through this approach, the connection between the WOFOST model and the GRU model was effectively established. As a crucial state variable in WOFOST, LAI signifies the photosynthetic capability of crops and can effectively characterize the potential yield (Huang et al., 2015). Accounting for the computational efficiency of the model in large areas, the average value of LAI was used as the input feature of GRU. To better capture the growth dynamics of soybean, the mean LAI at vegetative (from emergence to flowering) and reproductive (from flowering to maturity) growth period in the soybean growth dataset were calculated, represented as $LAI_{mean1}$ and $LAI_{mean2}$, respectively. The two stage LAI values derived from the simulations, **serving as temporal input features,** were then combined with model simulated soybean yield in knowledge base, **serving as model output,** to train the GRU model. The simulated soybean growth dataset was splits into training and testing datasets using 10-fold cross-validation **for model training.** The hyperparameters of GRU was optimized using a grid search method (Açikkar, 2024). The root mean squared error (RMSE, Eq. (5)) was applied to assess the predictive performance of different set of hyperparameters. After optimization of each fold, the hyperparameters that yielded the smallest predictive error were selected as the optimal ones. **The trained GRU model was then coupled with remote sensing data for large-scale soybean yield mapping by inputting the feature variables inversed with remote sensing data.***

11. Why is MODIS data mentioned again in Line 315? MODIS data was not mentioned in the data collection section.

**Reply:** Thank you for pointing out this. This is a writing error, and we apologize for the confusion. The term "MODIS data" in Line 315 actually refers to the MODIS LAI product (MCD15A3H), which was used in the study. We have corrected this in the revised manuscript to avoid any misunderstanding (Line 317 - 322).

*In this framework, the MODIS LAI data, MCD15A3H, was used for intercalibration. Due to its higher temporal resolution and broader image coverage, MCD15A3H generally provided more continuous estimation results. For correction, yield maps were also generated using MODIS LAI products. We utilized the estimation results from MODIS LAI to calibrate the mean yield for each Sentinel-2 tile. The difference between the mean value of the yield derived from MODIS LAI for the region cover the tile and the initial Sentinel-2 estimations was then added to the yield values from Sentinel-2.*

12. As shown in Figures 6, 7, and A2, the model simulation accuracy is below industry standards.

**Reply:** Thank you for pointing this out.

We have made clarifications in the revised manuscript. While the accuracy of our

model may appear to be below industry standards in Figures 6, 7, and A2, we have provided a more comprehensive evaluation of its performance in comparison to other studies. The comparison between the performance of our study with that of other studies at multi-scales (field and municipal scale) showed that our method outperformed existing approaches in terms of accuracy. Moreover, we compared our results with soybean yield datasets from other countries with similar resolution. Results showed that our dataset demonstrated superior accuracy. We acknowledge that some studies based on UAV and RGB data have reported higher accuracy for soybean yield estimation. However, these methods are limited by challenges related to data acquisition and high costs, making them suitable only for individual plant or field scale analysis. This limits their applicability for large-scale studies.

The primary goal of this study is to provide a hybrid modeling approach that enables rapid and large-scale soybean yield estimation. The method we produced balances computational efficiency, accuracy, and high resolution, making it suitable for regional-scale and field scale applications. This approach represents a practical solution for large-scale yield estimation despite the lower accuracy compared to some high-cost methods.

Details of revision could be found in Line 469 – 478.

*Accurate monitoring of soybean yield is crucial for food policy decision-making and security assessment. Previous studies have primarily focused on the impact of various factors (e.g., climate) on soybean yield (Guo et al., 2022; Zhao et al., 2023a). To our knowledge, high-resolution soybean yield dataset is currently unavailable in the main production regions of China. The study combined crop growth model with deep learning to construct a hybrid model driven by data and knowledge simultaneously for soybean yield estimation. The model retained its data mining capabilities while incorporating mechanistic constraints, thereby enhancing the model's interpretability and transferability. Accuracy verification based on in-situ and statistical data showed that the NortheastChinaSoybeanYield20m generated in this study accurately estimated soybean yield at both field and regional scales (Fig. 5 and 6). **Compared to existing studies combining remote sensing with process-based model for soybean yield estimation (e.g., Baup et al., (2015), reporting estimation errors of 2 -18% ), our method achieved comparable ccuracy. Notably, our framework outperformed existing approaches at both field and regional scales. At the field scales, the assessments obtained RMSE of 287.44 kg ha$^{-1}$ that surpass the RMSE of 400.946 kg ha$^{-1}$ reported by Ren et al., (2023a). At the municipal scale, the accuracy was RMSE of 272.36 kg ha$^{-1}$ that was lower than RMSE of 420.00 kg ha$^{-1}$ reported by Von Bloh et al., (2023). Additionally, the NortheastChinaSoybeanYield20m dataset demonstrated superior accuracy compared to similar resolution soybean yield datasets from other countries (Song et al., 2022). While UAV -based RGB data achieved higher point-scale accuracy (Li et al., 2021, 2024), their reliance on costly, localized data acquisition limits scalability. The method developed in this study strikes a balance between computational efficiency, spatial resolution and accuracy, offering a practical solution for large-scale yield estimation.***

13. By the way, Line 240:" 3.1.2 Multi-scenarios crop simulations", author said:" The four different types of model parameters were arranged and combined to generate various simulation scenarios". Where could I read the scenario settings and the results of this part in the manuscript?

Reply: Thanks for your suggestion.

In the revised version, we have provided a more detailed description of Section 3.1.2: Multi-scenario crop simulations. The study simulated different soybean growth scenarios by fully configuring four input parameters of the WOFOST model: meteorological parameters, soil parameters, crop-specific parameters, and agro-management parameters. The meteorological parameters were derived from observational data collected over 42 years from 51 meteorological stations, while 4 soil types, 5 crop varieties, and 4 agro-managements were defined. By combining different parameter types (similar to a lookup table approach), we inputted these parameter combinations into the WOFOST model to simulate various scenarios. To more clearly describe the scenario simulation process, we have added a Table 3 in the revision, which outlines the scenario settings in detail. Furthermore, we have corrected a numerical error in the revised version. The total number of simulated scenarios generated by the parameter combinations exceeds 170,000, rather than 80,000 as previously stated. We sincerely apologize for the oversight in the earlier version.

We have conducted a thorough review of the revised version to ensure that similar errors have been avoided and the accuracy of the content is maintained.

*After parameter preparation, a soybean growth dataset was constructed through model simulations which accounted for the multi-scenarios in agricultural production. The four different types of model parameters (meteorological parameters, soil parameters, crop-specific parameters and agro-management parameters) were arranged and combined to generate various simulation scenarios (Table 4). The scenarios were then put into the model for simulation. Finally, a dataset containing more than 170,000 (51×42×4×5×4) available simulations were generated.*

**Table 3 Scenarios for WOFOST simulations**

| Parameters | Number of categories | Details |
| --- | --- | --- |
| Meteorological parameters | 51×42 | Meteorological data from 51 stations over 42 years (1980 – 2021) |
| Soil parameters | 4 | Sandy loam, light loam, medium loam and heavy loam |
| Crop-specific parameters | 5 | Early maturity, medium-early maturity, intermediate maturity, medium-late maturity and late maturity |
| Agro-management parameters | 4 | Four planting dates 20 April, 30 April, 10 May, and 20 May |

---

## Author Comment (AC2)

**Responses to the comments of Referee #1**

Article ID: essd-2024-586

**Title:** NortheastChinaSoybeanYield20m: an annual soybean yield dataset at 20 m in Northeast China from 2019 to 2023

Authors: Jingyuan Xu, Xin Du, Taifeng Dong, Qiangzi Li, Yuan Zhang, Hongyan Wang, Jing Xiao, Jiashu Zhang, Yunqi Shen, Yong Dong

Dear Reviewer,

Thank you very much for your thorough review and constructive feedback on our manuscript. We have carefully addressed each comment and suggestion to refine our work, enhance its clarity and strengthen its scientific contribution. The key revisions include:

(1) The description of the data has been thoroughly revised to eliminate any ambiguity and prevent potential misinterpretations by users, ensuring greater clarity and accuracy in the presentation.

(2) We have added more details of the method to enhance the scientific rigor of the article.

(3) Many paragraphs, sentences, and figures have been revised to improve readability, conciseness, and clarity.

The detailed point-to-point responses are as follows. Texts in black are the reviewer's comments; those in blue are our responses to the reviewer's comments; and those in *red and italics* are the revised texts appeared in the revised manuscript.

I am very familiar with the WOFOST model and the dataset used by the author. It is not a good simulation project, not only because the simulation accuracy did not meet industry standards, but also because the author withheld many critical details and settings of the WOFOST in the manuscript, which makes it difficult for me to assess the rationality and scientific validity of the simulation. *Earth System Science Data*, as the name suggests, focuses on the application of datasets, but the author's professionalism in describing and processing the dataset is not good. Moreover, the description of CRU is severely inadequate. After reading the entire manuscript, I still do not understand the role of the CRU used by the author in this study.

**Reply:** We greatly appreciate the thoughtful and constructive feedback provided by the reviewer. We have carefully considered all the comments and have made substantial revisions to the manuscript to address the concerns raised, especially in the areas of dataset descriptions, simulation details and model contributions.

(1) Clarification of WOFOST Model and Dataset Details: We acknowledge the concern regarding the lack of critical details about the WOFOST model and its settings. In the revised manuscript, we have made significant improvements in the description of the dataset used, especially for the soil data (Section 2.2.3) and statistical data (Section 2.2.6). Specifically, we have provided more detailed information on the model's input parameters, for example, a more detailed explanation was provided on how soil parameter values were obtained, complete crop parameter settings were included (Table A1), and the production scenarios considered when setting agro-management parameters were discussed. These changes aim to improve the transparency and scientific rigor of our simulation process, allowing for a better assessment of the model's rationality and scientific validity.

(2) **Revision of GRU Model Description:** A major revision was made in the section describing the GRU model. In the previous version, the role and contribution of the GRU model were not sufficiently highlighted. In the revised manuscript, we have provided a more detailed and comprehensive description of how the GRU model was integrated with the WOFOST model. Specifically, we have clarified how it interacts with the outputs of the WOFOST model and how the GRU model utilizes remote sensing data as input to estimate soybean yield (Section 3.2). The connection between the two models is now described in greater detail (Figure 2), outlining the specific features that were used in the GRU model training, as well as how to estimate soybean yield using remote sensing data.

(3) Yield Estimation Accuracy Comparison: In response to your comments on the accuracy of the yield estimation, we have revised the section to include more detailed comparisons with other studies at various spatial scales. We have compared the performance of our method with studies conducted at both field and municipal scales, highlighting the improved accuracy of our estimates. Additionally, we have expanded the discussion on the comparison of our dataset with soybean yield datasets from other countries to better demonstrate the higher precision of our estimates. This comparison includes a detailed analysis of RMSE and  $R^2$  values, supporting the claim

that our method provides reliable and accurate yield estimates (Section 5.2).

We believe these revisions address the concerns raised and significantly enhance the quality and clarity of the manuscript. We sincerely hope that the updated version meets your expectations and are confident that the improvements made will contribute to a better understanding of our approach.

Below, we provide a detailed point-by-point response to address the specific concerns you raised:

1. The study spanned from 2019 to 2023, but the sampling data was only from 2022 and 2023 (Fig.1). The author should explain this issue in the text.

**Reply:** Thank you very much for the suggestion. In the revised manuscript, we have clarified that although the study spanned from 2019 to 2023, field observations were not conducted from 2019 to 2021 due to resource limitations. As a result, the sampling data for this study were collected only in 2022 and 2023. (Section 2.2.1)

Due to limitations of resources and personnel, in-situ measurements were not available during the earlier years (from 2019 to 2021). Field-scale yield data was separately collected through field investigation in September 2022 and 2023.

2. The soil data should be described in more detail, for example, which soil parameters were used in this study.

**Reply:** Thanks for your suggestion. We have provided a more detailed description of the soil data used in this study. For our study, we did not use the soil attribute data (such as chemical characteristics) in this analysis, focusing only on the spatial distribution of soil types to characterize the soil types in the study region. This clarification has been added to the revised manuscript (Section 2.2.3).

Soil data was obtained from the 1:1000,000 Chinese soil database, established by the Institute of Soil Science, Chinese Academy of Sciences (Shi et al., 2004). The dataset consisted of two parts: soil spatial data (digital soil maps) and soil attribute data. In this study, the 1:1000,000 soil spatial data was obtained. The spatial database was developed by digitizing, mosaicking, and reassembling sheets from the 1:1,000,000 Soil Map of the People's Republic of China (National Soil Survey Office, 1995), with the Genetic Soil Classification of China (GSCC) soil families as the fundamental mapping units. The final dataset includes 909 soil types and over 94,000 polygons. The dataset was utilized to determine the dominant soil types within the study area, serving as the basis for assigning soil parameter settings according to literatures.

3. The author used statistical data from 1980 to 2022, but the study's time scale is

from 2019 to 2023. This is confusing for the readers. Please provide an explanation.

**Reply:** Thanks for your suggestion. We have clarified the use of statistical data in the revision. In this study, the statistical data served two main purposes. Firstly, to ensure that the multi-scenarios soybean growth dataset, which was constructed in this study, accurately represented a wide range of soybean production conditions, the statistical data from 1980 to 2022 were all used to validate the reasonableness of the model simulations. Secondly, statistical data from 2019 to 2022 were specifically applied to evaluate accuracy of soybean yield estimation at the regional scale. This distinction has been clearly explained in the revised manuscript to avoid any confusion for the readers (Section 2.2.6).

Crop yield records (1980-2022) were obtained from the Statistical Yearbooks published by the Statistic Bureau of Heilongjiang (http://tjj.hlj.gov.cn), Jilin (http://tjj.jl.gov.cn), Liaoning (https://tjj.ln.gov.cn) and Inner Mongolia Autonomous Region (https://tj.nmg.gov.cn) to validate the crop yield estimates. Because the 2022 Statistical Yearbook was not fully released, yield records for that year cover only a subset of cities. The statistical data served two main purposes, model simulation validation and regional-scale accuracy evaluation in this study. To ensure the multi-scenario soybean growth dataset capture the full range of production conditions that across multi-years meteorological data, various soil types, multiple soybean varieties and different agro-managements, the yield records from 1980 to 2022 along with published yield data and field samples were used to assess the reasonableness of simulated yields. For the spatial validation, regionally aggregated statistical yield data (2019 – 2022) were applied to evaluate the accuracy of the hybrid framework at municipal and provincial scales.

4. The technology roadmap that needs improvement. 1) The author mentioned agromanagement data in Figure 2, but it is not mentioned in Section 2.2 Data collections. 2) The sampling data mentioned in the Data collections section is not reflected in the figures, as well as meteorological data from National Meteorological Information Center. 3) The method of combining remote sensing data and model output through GRU is described too simplistically. 4) The author allocates a large proportion of the figures to how WOFOST conducts simulations, but this is not the focus of this study. The focus of this study should be on how to use models and remote sensing coupled for yield estimation, just as the author introduces in the research objective: "Designing a hybrid model coupling crop growth model and deep learning model for soybean yield estimation." The technology roadmap should more detailed display the research focus.

**Reply:** Thank you for pointing this out. We have made substantial revisions to improve the technology roadmap, addressing the points raised:

(1) The study primarily simulated different soybean agro-management scenarios by

setting different planting dates. The agro-management data was collected alongside in-situ measurements. We have now included an explanation of the agro-management data collection process in Section 2.2.1 to ensure consistency between the figure and the text.

Due to limitations of resources and personnel, in-situ measurements were not available during the earlier years (from 2019 to 2021). Field-scale yield data was separately collected through field investigation in September 2022 and 2023. In each year, a total of 21 and 18 sample plots were selected, respectively (Fig. 1). Within each sample plot that was around 100  $m \times 100$  m in area, nine quadrats with area of 1  $m \times 1$  m were selected randomly for destructive sampling of yield in soybean. The central location of each quadrat was recorded using a GPS device with accuracy of 1 m. The harvested beans were then oven-dried about 72 hours in Hailun Agricultural Ecology Experimental Station, Chinese Academy of Sciences to determine the yield. Finally, the average yield for the selected nine quadrats represents the soybean yield of the sample plot. In addition, soybean planting dates for different regions were collected through field surveys, providing agro-management data for this study.

- (2) The sampling data were used to validate soybean yield estimation accuracy at the field scale, while the meteorological data from the National Meteorological Information Center were used as input for the WOFOST model to provide essential weather parameters. Both of these datasets are represented in the revised figure (field measured samples and meteorological station data, respectively), and their roles have been explicitly described in the text.
- (3) We have revised the flowchart to better represent the hybrid model's structure. The updated version now clearly distinguishes two main components: the first part describes the construction of the hybrid model, combining WOFOST and GRU for yield estimation, while the second part focuses on how remote sensing data are used in conjunction with this hybrid model to estimate soybean yield at the regional scale. We have provided more detail in the flowchart regarding how remote sensing data are integrated as inputs into the GRU model and how they contribute to spatial yield predictions.
- (4) We agree that the focus of the study should be on the coupling of remote sensing data with models for yield estimation, as stated in the research objectives. Therefore, the updated technology roadmap places greater emphasis on this aspect, reducing the proportion dedicated to the WOFOST model and highlighting how the hybrid model (WOFOST + GRU) is used for yield estimation. This revision better aligns the roadmap with the research focus and research objectives, as well as the methodology described in the manuscript.

---

## Author Comment (AC3)

**Responses to the comments of Referee #2**

Article ID: essd-2024-586

**Title:** NortheastChinaSoybeanYield20m: an annual soybean yield dataset at 20 m in Northeast China from 2019 to 2023

Authors: Jingyuan Xu, Xin Du, Taifeng Dong, Qiangzi Li, Yuan Zhang, Hongyan Wang, Jing Xiao, Jiashu Zhang, Yunqi Shen, Yong Dong

Dear Reviewer,

Thank you very much for your thorough review and constructive feedback on our manuscript. We have carefully addressed each comment and suggestion to refine our work, enhance its clarity and strengthen its scientific contribution. The key revisions include:

(1) **Strengthened the Introduction section.** We have restructured the introduction to better contextualize the critical issues on coupling data-driven and knowledge-drive methods for crop yield estimation. The revised edits now explicitly highlight the limitations of the existing models (e.g., coarse resolution, and over-reliance on ground data).

(2) Expanded details in data processing. We expanded the Data collection section to include details on data processing procedures especially for the meteorological and satellite imagery data.

(3) Enhanced interpretation of results. We strengthened the Results and Discussion sections by analyzing yield estimation uncertainty across different scales and discussing key sources of error, providing deeper insights into model performance and study implications.

(4) Quantified MODIS-Sentinel-2 Comparison in yield estimation. We have added a new subsection in the Discussion section on quantitative comparison of the performance of MODIS LAI and Sentinel-2 data in yield estimation.

The detailed point-to-point responses are as follows. Texts in black are the reviewer's comments; those in blue are our responses to the reviewer's comments; and those in *red and italics* are the revised texts appeared in the revised manuscript.

The overall structure of the article is clear and logically organized. The research demonstrates innovation by integrating crop growth models with deep learning algorithms for soybean yield estimation, representing a promising direction in agricultural remote sensing. The research objectives are well-defined, aiming to address existing limitations in soybean yield data (insufficient spatial resolution and reliance on ground observations), thereby supporting optimized soybean production distribution and agricultural decision-making.

**Reply:** Thank you for your positive feedback and recognition of our work. In the revision, we have carefully addressed your thoughtful comments and suggestions to improve our manuscript.

**Specific Comments:**

1 Introduction: The section comprehensively highlights soybean's global food security significance and limitations of current yield estimation methods, establishing a solid research rationale. However, the comparative discussion of data-driven and knowledge-driven methods could be more concise to better emphasize core issues and proposed solutions. Additionally, enhancing explanations of environmental factors' mechanisms (e.g., how climatic conditions affect growth cycles and photosynthesis, or how soil properties constrain nutrient uptake and water retention) would provide a more systematic understanding of key yield determinants and their interactions.

**Reply:** Thank you for your valuable comments.

- (1) In the revised revision, we have refined the statements on the advantages and limitations of existing methods, placing greater emphasis on our proposed method.
- (2) We discussed the impact of environmental factors (climate conditions and soil properties) on crop growth in the introduction. Specifically, we clarified the limitations of data-driven methods in accounting for environmental factors, and highlighted the strengths of knowledge-driven models incorporating these influences.

Revisions can be found in Section 1 Introduction. Below is a part of the revision for your reference:

Data-driven methods leverage satellite-derived variables such as leaf area index (LAI), fraction of absorbed photosynthetically active radiation (FAPAR), and vegetation indices (VIs) to establish linear or nonlinear relationships with measured crop yield (Ang et al., 2022; Xie et al., 2019). Machine learning algorithms such as Random Forest (RF), and Artificial Neural Networks (ANN), due to their ability to process large dataset and model complex nonlinear interactions, have been widely applied in crop yield estimations (Pang et al., 2022; Tian et al., 2021; Yildirim et al., 2022). These methods can extract effective information from multi-source structured

or unstructured data without manual intervention. However, they are heavily reliant on extensive ground-truth training data, which is challenging to collect over large areas and high time intervals (Cao et al., 2021). Additionally, these models often overlook the impacts of environmental factors on crop growth, such as the influence of early-season soil moisture on root establishment or the effect of high temperatures during flowering on pod set, and are lack of interpretability, as they cannot explain the causal relationship between input features and outputs, leading to poor spatial-temporal generalization (Gevaert, 2022).

In contrast, knowledge-driven crop growth models simulate crop development from sowing to harvest based on agronomic mechanisms (Kaur and Singh, 2020). Common model types include light-use efficiency models (e.g., SAFY (Duchemin et al., 2008)), soil-driven models (e.g., AquaCrop (Steduto et al., 2009)), and atmosphericdriven models (e.g., WOFOST (Diepen et al., 1989)). These models integrate environmental factors (e.g., climate conditions and soil characteristics) with crop physiological processes (Gaso et al., 2024). Climate variables like temperature, precipitation, and solar radiation are critical in regulating essential physiological processes such as photosynthesis, respiration and transpiration, which influence the rate and duration of crop growth stages (Misaal et al., 2023). Climate anomalies during specific growth stages may disrupt biochemical processes, ultimately affecting yield formation. Similarly, soil properties influence crop productivity by regulating water retention, aeration, and nutrient uptake (Muhuri et al., 2023). Despite their mechanistic rigor, applications of crop models over large area are typically constrained by (1) insufficient spatial-temporal input data, and (2) parameter uncertainty, which can propagate errors into yield estimations (Dokoohaki et al., 2021). To overcome these challenges, data assimilation techniques to integrate remote sensing observations (e.g., LAI) into crop growth models have been developed to enhance spatial representativity (Huang et al., 2024). However, high resolution remote sensing data drastically increases computational cost, limiting the scalability of these approaches for regional or national mappings efforts (Huang et al., 2019).

Given the limitations above, integrating data-driven and knowledge-driven models has emerged as a critical strategy to enhance spatial-temporal generalization and mitigate sparse training data challenges in crop yield estimations. Hybrid frameworks coupling crop growth model with machine learning algorithm, such as those proposed and evaluated by Ren et al., (2023b) and Xie and Huang, (2021), are gaining tractions.

2 Data Collection: The dataset (field measurements, meteorological/soil data, satellite imagery, crop distribution maps, and statistics) is comprehensive and representative. However, data processing steps (e.g., meteorological data interpolation, satellite image preprocessing) require more detailed technical descriptions to improve reproducibility. Furthermore, explicit clarification is needed regarding spatial alignment and scale conversion methods employed for integrating multi-resolution datasets.

**Reply:** Thanks for your suggestion.**

We have carefully revised the Data Collection section to provide a more detailed description of the data processing procedures, particularly for the meteorological and satellite imagery data.

- (1) In the revised version, we clarified the purposes and preprocessing steps for the two climate datasets (meteorological station data and climate reanalysis data) used in the study. We detailed the procedures used to address missing values and outliers in the meteorological station data. We described the resampling method employed to align the spatial resolution of ERA5 product with that of satellite imagery. (Section 2.2.2)
- (2) Moreover, we expanded the description of data processing for the two satellite datasets (Sentinel-2 and MODIS LAI). We clarified that since yield maps were generated independently from each dataset for subsequent yield bias correction, we only performed reprojection to spatially align the imagery. (Section 2.2.4)
- 2.2.2 Meteorological data

In this study, two different climate datasets were used.

The meteorological station data used in this study came from the meteorological stations of the National Meteorological Information Center (http://data.cma.cn). There are 238 meteorological stations within the study area. Here 51 of the meteorological stations that located within 1 km buffer zone of the soybean cultivation areas were selected (Fig. 1). The meteorological datasets generally include insolation duration (h), minimum temperature (°C), maximum temperature (°C), daily average temperature (°C), average water vapor pressure (kPa), average wind speed (m sec-1), precipitation (mm) and snow-depth (cm). Observed data from 1980 to 2021 of the 51 selected stations were collected. Missing values and outliers in the data were filtered out. The data were then directly used for setting input climate parameters of the WOFOST model to drive simulations.

The climate reanalysis data was obtained from the ERA5-land Daily Aggregated -ECMWF Climate Reanalysis Product. The data was only used to calculate soybean phenology for preparation of yield estimations. It was a global climate reanalysis product that provides continuous climate data at a resolution of  $0.1^{\circ} \times 0.1^{\circ}$  (e.g., air temperature and atmospheric pressure) starting from 1950. The daily aggregated air temperature data at 2 m above the surface of land measured in kelvin (K) during the soybean growth periods from 2019 to 2023 was collected in this study from the Google Earth Engine (http://earthengine.google.com). The product was resampled to 20 m using bilinear interpolation model to match with the resolution of satellite imagery data.

**2.2.4 Satellite imagery data**

Two satellite data including: 1) Sentinel-2 Multi-Spectral Instrument (MSI) Level - 2A Surface reflectance product (10 – 60 m spatial resolution, 5-day revisit), and 2) the Moderate Resolution Imaging Spectroradiometer (MODIS) Leaf Area Index (LAI) / Fraction of Photosynthetically Active Radiation (FPAR) Level 4 product (MCD15A3H, v061, 500 m spatial resolution, 4-day period) were used to generate yield maps. All data spanning soybean growth periods (2019 – 2023) were accessed and pre-processed via the Google Earth Engine (GEE, http://earthengine.google.com).

The MSI aboard Sentinel-2A/B satellites provides 10 m (visible and near-infrared bands), 20 m (red-edge and shortwave infrared bands) and 60 m (atmospheric bands) bands at 5-day revisit. The Level-2A data, which are geometrically and atmospherically corrected via the Sen2Cor, were masked for clouds and shadows using the Quality Assurance (QA) band. The 60 m band was excluded due to their low spatial resolution and limited relevance for yield estimation and the 10 m (B2: Blue, B3: Green, B4: Red, B8: Near-Infrared) and 20 m (B5–B7: Red-edge, B8A: Near-Infrared, B11–B12: Shortwave Infrared) bands were retained. To harmonize spatial resolution, the 10 m bands were resampled to 20 m using bilinear interpolation model.

The MODIS MCD15A3H (Collection 6.1, Level 4) provides 4-day composite LAI and FAPAR at 500 m derived from Terra and Aqua satellite sensors LAI/FAPAR are primarily inverted via a 3D radiative transfer model-based look-up-table (LUT) algorithm (Knyazikhin et al., 2018). When the primary algorithm fails, they are estimated using an empirical NDVI-LAI model. The LAI data was similarly reprojected to WGS -84 to ensure spatial alignment with Sentinel-2 imagery. These coarse-resolution LAI data were used to generate 500 m yield maps. The coarseresolution yield maps were then used to bias-correct the 20 m Sentinel-2 yield maps, improving their regional consistency. Details about the bias correction are present in following 3.3.2 Section.

3 Results: Results are effectively visualized through figures/tables demonstrating WOFOST model simulations, multi-scale estimation accuracy, and spatial yield patterns. The analysis appropriately discusses model accuracy, stability, and spatiotemporal pattern recognition capabilities. However, deeper interpretation of anomalies (e.g., regional/yearly estimation errors) is needed. Notably, the systematic overestimation in field-scale validation suggests potential model biases (e.g., systematic errors or overfitting), warranting further investigation.

**Reply:** Thanks for your suggestion.**

In Result Section of the revision, we have expanded our analysis of uncertainty in soybean yield estimation at the field (Section 4.2) and regional (Section 4.3) scales. In the discussion section, we conducted a more detailed assessment of the model's estimation errors across different scales, regions, and years. The interpretation of the results is framed around two key aspects: (1) systematic errors intrinsic to WOFOST model simulations, and (2) overfitting tendences of the GRU model. On this bias, we

further discussed the limitations of the current study and suggested the directions for future research. (Section 5.3)

**4.2 Yield estimation at field scale**

The field-scale performance of NortheastChinaSoybeanYield20m was validated against in-situ measurement from 2022 and 2023, demonstrating strong accuracy in capturing spatial yield variability (Fig. 5). The estimated yields showed strong agreement with observed yield, with  $R^2 > 0.65$  (p < 0.01). The error-bars indicated more consistent performance in fields with uniform yields, while higher uncertainties appear in fields with larger estimation deviations. Overall accuracy across both years reached 0.73 in  $R^2$  (p < 0.01), 287.44 kg ha-1 in RMSE and 10.02 % in MRE (Fig. A2). Notably, higher accuracy in 2023 with RMSE of 271.07 kg ha-1 and MRE of 8.57 % (Fig. 5b) was achieved. The results indicated that the dataset well captured the spatial variation of soybean yield.

**4.3.1 Variability of accuracy through years**

The NortheastChinaSoybeanYield20m was validated at the municipal scale (2019 to 2022) by aggregating yield maps to match statistical data (Fig. 6). Compared to the field-scale validation, the municipal-scale estimates exhibited greater uncertainty, likely reflecting increased heterogeneity of soybean yields over larger areas. The estimates maintained stable interannual performance, with correlation between estimated and statistical yields consistently exceeding 0.60 (p < 0.01). The overall accuracy, pooled across 2019- 2022, for municipal-scale achieved  $R^2 = 0.62$  (p < 0.01), RMSE = 272.36 kg ha-1, and MRE = 12.08 % (Fig. 11a). Annual accuracy metrics ranged from 221.69 kg ha-1 to 310.66 kg ha-1 for RMSE and from 8.24 % to 14.40 % for MRE, with the 2022 year achieving the highest accuracy (MRE < 10%, Fig. 6d).

**5.3 Limitations and future developments**

In this study, a multi-scenario soybean growth dataset was developed by simulating various combinations input parameters within the WOFOST model. These diverse scenarios were designed to reflect different environmental and management conditions, ultimately serving as training data for the yield estimation model. One advantage of the model is its scalability, it can be readily applied to other regions and countries that lack sufficient ground observation data, such as parts of Africa and India, thus offering a promising tool for global agricultural monitoring.

However, the validation results revealed some notable limitations. Specifically, the model exhibited a tendency to produce large uncertainty in low- or high- yielding areas, introducing error into the overall yield estimation (Fig. 5 and 6). This pattern suggests a systematic bias in the model's predictions, particularly in regions with extreme yield values. Additionally, spatial analysis showed that estimation errors were more pronounced in the northern region, where is characterized by complex terrain,

compared to the relatively flat central region (Fig. 7). These discrepancies highlight the need to refine parameterization for extreme yield conditions and integrate higherresolution environmental drivers (e.g., terrain, localized weather).

On the one hand, the estimation errors may be attributed to the inherent limitations of the WOFOST model. As a process-based model, WOFOST simplifies its calculations for simulating physiological processes, which can hinder its ability to fully replicate the complex realities of soybean in the field. Factors, such as pest infestations, diseases, and abiotic stresses are either oversimplified or excluded (Gaso et al., 2024). These omissions can lead to systematic simulation errors, particularly under stress conditions that significantly affect crop yield. Moreover, the parameterization of the WOFOST model in this study purely relied on values from literature and existing dataset rather than local optimization. As a result, local variability because of farming practices, soil properties, and environmental conditions may not have been adequately captured. This lacks local optimization likely result in higher estimation error, especially in complex landscapes with spare ground observations. To address these issues, future works incorporating fieldspecific parameters or advanced data assimilation techniques could help reduce bias and improve model accuracy across heterogeneous landscapes. Given the spatial variability in soybean growth within the study area, constructing ecological zones based on factors like climate, elevation, and management practices might provide a more targeted model approach. For instance, Huang et al., (2023) defined the ecological zones through using Theissen polygons derived from meteorological station locations. This zoning strategy could enhance the representativeness of the training data and reduce yield estimation uncertainties.

On the other hand, the estimation errors may stem from the overfitting of the **GRU model.** The GRU was trained on the multi-scenarios simulated dataset, a large number of simulations that included all available combinations (e.g., all meteorological data), which introduced a significant amount of redundant information. The redundancy not only potentially reduce the dataset's representativeness, but also increase the computational burden during model training. As a result, the trained GRU model may have become overly turned to specific temporal patterns in certain years, limiting its ability to generalize to other time period or regions with different growth conditions. This overfitting effect might result in large yield estimation errors across different years and regions, particularly in areas where soybean yields deviated significantly from the norm. To address these issues, refining the structure and composition of the training dataset, and removing redundant information would enhance the diversity and quality of the training inputs. One potential approach to reduce redundancy is through spatiotemporal clustering of various environmental (e.g., meteorological station data), which could filter out stations with highly similar information. Moreover, monitoring the validation error throughout the training process, and implementing regularization techniques (e.g., L2 weight regularization) could help to prevent overfitting and improve the GRU model's generalization capability, leading to improve soybean estimation across varying conditions...

4 Discussion: When discussing MODIS-Sentinel-2 complementarity, quantitative comparisons of their performance under varying conditions (weather/vegetation coverage) would strengthen data selection guidance. Future research directions could be expanded by aligning with emerging trends (e.g., integration with IoT/blockchain technologies, precision agriculture applications), thereby enhancing both theoretical depth and practical relevance for agricultural challenges.

**Reply:** Thanks for your suggestion.**

- (1) We compared the yield estimation performance of MODIS and Sentinel-2 under different conditions in the Discussion section (Section 5.1). Specifically, In the revised manuscript, we established 10 km grids across the study area and calculated soybean coverage of each cell. We then randomly selected three representative grid cells, corresponding to coverage thresholds of <25%, >50%, and >75%. For each selected grid cell, we extracted Sentinel-2 yield maps and MODIS LAI yield maps from 2019 to 2023 to facilitate a systematic comparison. Accordingly, the Figure 13 has been updated to quantitatively illustrate the differences between the datasets.
- (2) Regarding future research directions, we have expanded our discussion on the future directions of research to explore the integration of emerging technologies such as IoT, blockchain, and precision agriculture with machine learning and biophysical models. Revisions can be found in Section 5.3.

This study generated soybean yield estimates using both MODIS LAI (500 m) products and S2 derived LAI (20 m) data. Over 2019 – 2022, the MODIS-based estimates achieved an overall  $R^2$  of 0.58 (p < 0.01), an RMSE of 272.36 kg ha-1 and an MRE of 12.08 % (Fig. 11b), slightly lower than the Sentinel-2 based results (Fig. 11a). The uncertainty of MODIS based estimates was higher than that the Sentinel-2 based estimates, likely reflecting MODIS's coarser resolution. However, the Sentinel-2 based estimates exhibit inherent seaming effects caused by cloud-affected tile edges. We additionally used MODIS LAI to bias-correct Sentinel 2 yield maps, effectively minimizing the striping ("seaming") effects in the 20 m products (Fig. 9), while preserving pixel-level detail through tile-based calibration (Fig. 13). Despite difference in spatial resolution, both MODIS and Sentinel-2 satellite data demonstrated comparable ability to capture spatiotemporal variation in soybean yield (Fig. 12), achieving correlations with statistical data > 0.55 and overall errors < 13 % across all years.

In practical applications, balancing both temporal and spatial resolution is critical for achieving robust yield prediction results (Azzari et al., 2017). Figure 13 compares the Sentinel-2 yield maps and the MODIS LAI yield maps within a 10 km grid under different soybean coverage. Thanks to 4-day revisit, MODIS LAI provides more cloud-free observations during the critical growth stages, improving the reliability of two LAI metrics (LAImean1 and LAImean2). Its coarser spatial resolution

also accelerates spatial processing over large areas. However, Sentinel-2's finer more effectively resolves intra field yield heterogeneity (Fig. 13). MODIS-derived maps occasionally underestimated yields due to mixed pixels containing non-crop features (e.g., infrastructure), whereas Sentinel-2 minimized such errors.

While this study prioritized high-resolution mapping (using MODIS solely for Sentinel-2 seam correction), combing high spatial data (e.g., Sentinel 2 or UAV imagery) with high temporal frequency satellites (e.g., geostationary sensors or radar) could provide an optimal data source for crop yield modelling (Gao and Anderson, 2019; He et al., 2018).

---

## Author Comment (AC4)

**Responses to the comments of Referee #3**

**Article ID:** essd-2024-586
**Title:** NortheastChinaSoybeanYield20m: an annual soybean yield dataset at 20 m in Northeast China from 2019 to 2023
**Authors:** Jingyuan Xu, Xin Du, Taifeng Dong, Qiangzi Li, Yuan Zhang, Hongyan Wang, Jing Xiao, Jiashu Zhang, Yunqi Shen, Yong Dong

Dear Reviewer,

Thank you very much for your thorough review and constructive feedback on our manuscript. We have carefully addressed each comment and suggestion to refine our work, enhance its clarity and strengthen its scientific contribution. The key revisions include:

(1) The abstract was refined to emphasize the research goals, methodology, and key findings more clearly.
(2) The introduction was improved by better structuring the background information, and clearly stating the novelty of the proposed hybrid framework for soybean yield estimation.
(3) The clarity and presentation of figures were improved by enhancing the resolution and redesigning the layout.
(4) The Discussion and Conclusion section was revised to better highlight the advantages of the research and provide a more concise summary of the key findings, emphasizing the effectiveness of the proposed hybrid framework.

The detailed point-to-point responses are as follows. Texts in black are the reviewer's comments; those in blue are our responses to the reviewer's comments; and those in *red and italics* are the revised texts appeared in the revised manuscript.

This study presents a well-structured and logically organized framework for high-resolution soybean yield estimation. The combination of process-based modeling with deep learning offers a novel perspective for enhancing agricultural monitoring capabilities. The objectives are clearly articulated, with a strong focus on improving soybean yield data accuracy to support agricultural decision-making and production optimization. The methodological approach is rigorous, leveraging diverse production scenarios to train the GRU model and applying time-series Sentinel-2 data for large-scale yield estimation. The evaluation using in-situ measurements and government statistical data provides strong validation, and the reported accuracy metrics indicate reliable model performance across spatial and temporal scales. There are some suggestions as follows, which can be considered for further improvement of the manuscript.

Reply: Thank you for your thorough review and recognition of our work. We have carefully considered them in our revisions to enhance the quality and clarity of the manuscript.

The research is well-founded and presents significant innovations. However, the abstract and introduction sections could benefit from more professional and polished language to enhance readability and better highlight the study's contributions. Refining the writing style would improve clarity, strengthen the articulation of the research objectives, and more effectively emphasize the novelty of the proposed hybrid framework.

Reply: Thank you for your insightful comments and suggestions. In response to your valuable feedback, we have carefully refined the abstract and introduction sections to enhance the readability of the manuscript.

In the revised sections, we have improved the description of the technological background, providing a clearer discussion of data-driven and knowledge-driven approaches in crop yield estimation, along with their respective limitations. The revision ensures a more seamless transition into our research objectives and highlights the advantages of the proposed hybrid model for yield estimation. These modifications improve the overall coherence of the manuscript and better emphasize its scientific contributions.

***Abstract.** Accurate monitoring of crop yield is critical for ensuring food security. While various yield datasets covering Northeast China exist, they were produced at a coarse spatial resolution and remain inadequate for capturing small-scale spatial heterogeneity. Current yield estimation methods, such as machine learning models and the assimilation of remotely sensed biophysical variables into crop growth models, are heavily reliant on ground observations and computationally expensive. To address these limitations, we propose a hybrid framework that couples the World Food Studies Simulation Model (WOFOST) and a Gated Recurrent Unit (GRU) model to generate*

a high-resolution (20 m) soybean yield dataset in Northeast China from 2019 to 2023 (NortheastChindaSoybeanYield20m). First, to generate a comprehensive training dataset, WOFOST was employed to simulate diverse soybean growth scenarios by accounting for variations in climates, crop varieties, soil types and agro-managements practices. The GRU model was then trained to establish relationships between model simulated leaf area index (LAI) and soybean yield. The trained model was applied to estimate soybean yield in Northeast China using time-series LAI derived from Sentinel-2 at key growth stages. The accuracy of estimates was evaluated using in-situ measurements and government statistical data. The overall accuracy was 287.44 kg ha$^{-1}$ and 272.36 kg ha$^{-1}$ in the root mean squared error (RMSE) for field and regional scale, respectively. The model exhibited consistent interannual stability, with mean relative error (MRE) averaging 11.46 % and 7.94% at the municipal scale and the provincial scale, respectively. The dataset effectively captured spatiotemporal yield variability, offering potentials for optimizing soybean production, guiding precise agriculture practices, and informing agricultural policy. The NortheastChinaSoybeanYield20m dataset is publicly available at https://doi.org/10.5281/zenodo.14263103 (Xu et al., 2024).

**1 Introduction**

Soybean is a crucial crop for both food and oil production, providing more than a quarter of the world's edible protein (Graham and Vance, 2003). Global demand for soybean is projected to increase by 46 % by 2050, driven by rapid population growth (Falcon et al., 2022). As an major traded agricultural commodity, soybean production in key exporting nations has wide-reaching effects on international markets, and can significantly influence agricultural economies worldwide (Qiao et al., 2023). Notably, China is the world's largest consumer of soybeans (FAOSTAT, 2022), and its soybean demand relies heavily on international trade (Zhao et al., 2023). Consequently, accurate monitoring of soybean yield is vital for promoting sustainable agriculture, ensuring food security, and maintaining economic stability from regional to global scale. Moreover, effective yield monitoring and mapping supports farmers by informing field management practices, bolstering agricultural insurance and enhancing poverty alleviation initiatives (Zhuo et al., 2022).

Remote sensing data provides time-series observations for crop yield estimation across multiple scales (e.g., field, regional and national) (Dong et al., 2020; Hunt et al., 2019; Zhao et al., 2023b). Current methodologies for yield estimation can be broadly categorized as data-driven or knowledge-driven approaches.

Data-driven methods leverage satellite-derived variables such as leaf area index (LAI), fraction of absorbed photosynthetically active radiation (FAPAR), and vegetation indices (VIs) to establish linear or nonlinear relationships with measured crop yield (Ang et al., 2022; Xie et al., 2019). Machine learning algorithms such as Random Forest (RF), and Artificial Neural Networks (ANN), due to their ability to process large dataset and model complex nonlinear interactions, have been widely applied in crop yield estimations (Pang et al., 2022; Tian et al., 2021; Yildirim et al., 2022). These methods can extract effective information from multi-source structured

*or unstructured data without manual intervention. However, they are heavily reliant on extensive ground-truth training data, which is challenging to collect over large areas and high time intervals (Cao et al., 2021). Additionally, these models often overlook the impacts of environmental factors on crop growth, such as the influence of early-season soil moisture on root establishment or the effect of high temperatures during flowering on pod set, and are lack of interpretability, as they cannot explain the causal relationship between input features and outputs, leading to poor spatial-temporal generalization (Gevaert, 2022).*

*In contrast, knowledge-driven crop growth models simulate crop development from sowing to harvest based on agronomic mechanisms (Kaur and Singh, 2020). Common model types include light-use efficiency models (e.g., SAFY (Duchemin et al., 2008)), soil-driven models (e.g., AquaCrop (Steduto et al., 2009)), and atmospheric-driven models (e.g., WOFOST (Diepen et al., 1989)). These models integrate environmental factors (e.g., climate conditions and soil characteristics) with crop physiological processes (Gaso et al., 2024). Climate variables like temperature, precipitation, and solar radiation are critical in regulating essential physiological processes such as photosynthesis, respiration and transpiration, which influence the rate and duration of crop growth stages (Misaal et al., 2023). Climate anomalies during specific growth stages may disrupt biochemical processes, ultimately affecting yield formation. Similarly, soil properties influence crop productivity by regulating water retention, aeration, and nutrient uptake (Muhuri et al., 2023). Despite their mechanistic rigor, applications of crop models over large area are typically constrained by (1) insufficient spatial-temporal input data, and (2) parameter uncertainty, which can propagate errors into yield estimations (Dokoohaki et al., 2021). To overcome these challenges, data assimilation techniques to integrate remote sensing observations (e.g., LAI) into crop growth models have been developed to enhance spatial representativity (Huang et al., 2024). However, high resolution remote sensing data drastically increases computational cost, limiting the scalability of these approaches for regional or national mappings efforts (Huang et al., 2019).*

*Given the limitations above, integrating data-driven and knowledge-driven models has emerged as a critical strategy to enhance spatial-temporal generalization and mitigate sparse training data challenges in crop yield estimations. Hybrid frameworks coupling crop growth model with machine learning algorithm, such as those proposed and evaluated by Ren et al., (2023b) and Xie and Huang, (2021), are gaining tractions. These approaches utilized simulated outputs from crop growth models (e.g., meteorological, soil, crop physiological, and management factors) as inputs for machine learning, reducing reliance on limited ground observations Many studies have demonstrated hybrid methods are able to enhance yield estimation due to three benefits (Feng et al., 2020; Xie and Huang, 2021; Yang et al., 2021). The simulations from crop growth model can provide biophysical constraints to machine learning, ensuring agronomic plausibility. The crop growth models generate synthetic training datasets to address data scarcity. Finally, the machine learning improves the computational efficiency compared to traditional data assimilation techniques (Xie and Huang, 2021). However, exiting studies generally extracted input features (e.g.,*

*LAI, and soil moisture) across the entire growth cycle or on coarse temporal scales, increasing computational costs of model calculation and obscuring stage-specific physiological response (Pinke and Lövei, 2017; Wang et al., 2015). Additionally, while deep learning models, such as Long Short-Term Memory (LSTM) and GRU model excel at modelling temporal dependencies, their integration into hybrid frameworks have not been widely explored.*

*Critically, the primary soybean-producing regions of China lack a publicly available high-resolution yield dataset to analyse spatiotemporal production patterns, hindering precision agriculture and policy optimization. To address this, we developed a hybrid model coupling the World Food Studies (WOFOST) crop growth model with a GRU deep learning method to estimate soybean yield in Northeast China. The objectives include: (1) Design a hybrid framework integrating WOFOST-simulated growth scenarios with GRU-based temporal feature extraction; (2) Generate a high-resolution (20 m) soybean yield dataset in Northeast China (NortheastChinaSoybeanYield20m) from 2019 to 2023; (3) Evaluate the accuracy of the dataset across field, municipal, and provincial scales using in situ and statistical benchmarks. The WOFOST model first simulated a multi-scenario soybean growth (varying climate, soil, crop varieties and management conditions) to train the GRU model. The time series Sentinel-2 data, capturing soybean growth development, were then input into the GRU model to estimate yield. This approach prioritizes stage-specific physiological dynamics which balancing computational efficiency and spatial granularity, providing a critical advancement for scalable agricultural monitoring.*

Figure 1: where is the soybean classification map from? What is the accuracy?

**Reply:** Thank you for the comments.

The soybean map in Figure 1 was derived from existing study of Zhao et al., (2022) using an optimal identification feature (OIF) knowledge graph coupled with a moment-preserving segmentation method. The study classified maize, soybean, and rice in the Northeast China. The soybean distribution maps from 2019 to 2023 were collected in this study. The overall accuracy and the producer accuracy for maize, soybean and rice was higher than 90 % and 93 %, respectively, with a Kappa coefficient greater than 0.90.

In the revision, details on soybean classification maps were presented in Section 2.2.5. We have added additional information to Figure. 1, including the source of the classification map and classification accuracy.

*Figure 1: Location of the study area and the distribution of sample plots in two years (2022 and 2023) and selected meteorological stations. The soybean distribution map was obtained from Zhao et al., (2022) using a moment-preserving segmentation method, achieving an overall accuracy over 90% for soybean in 2023 (Details are provided in Section 2.2.5).*

*2.2.5 Crop distribution data*

*The soybean distribution maps for the study area (2019 – 2023) were obtained from Zhao et al., (2022), which employed a novel methodology for crop type identification. The study proposed an optimal identification feature (OIF) knowledge graph coupled with a moment-preserving segmentation method to classify crop types without ground-truth data. The method achieved overall accuracy above 90% and producer's accuracy exceeding 93% for maize, soybean and rice, with a Kappa coefficient greater than 0.90.*

*Zhao, L., Li, Q., Chang, Q., Shang, J., Du, X., Liu, J., and Dong, T.: In-season crop type identification using optimal feature knowledge graph, ISPRS Journal of Photogrammetry and Remote Sensing, 194, 250–266, https://doi.org/10.1016/j.isprsjprs.2022.10.017, 2022.*

Figure 5 appears blurry, which affects the clarity and readability of the presented data. I suggest organizing box plots and histograms as subfigures.

Reply: Thanks for your suggestion. In response, we have reorganized the histograms (a) and the box plots (b) as subfigures to present the data more effectively. We believe these adjustments improve the visualization and overall presentation of the results.

[Figure]

*Figure 2: (a) Histogram statistics of simulated soybean yield where the gray area in the histogram represents 95 % confidence intervals; (b) distribution of simulated soybean yield compared with other datasets where A represents simulated yield in this study (n = 171,360), B represents statistical yield from 1980 to 2022 (n = 961), C represents specific measurements from the literature (Chen et al., 2011; Fan et al., 2012; Liu et al., 2005, 2008; Liu and Herbert, 2002; Wang et al., 2020, 2024; Zheng and Zhang, 2021) (n = 138) and D represents measurements in 2022 and 2023 carried by this study (n = 39).*

The discussion on the advancements of the proposed method is embedded within the "Limitations and future developments" section. To better highlight the strengths of this study, I recommend extracting this content into a standalone subsection. This

would allow for a clearer and more structured presentation of the method's advantages, making it easier for readers to appreciate its contributions in comparison to existing approaches.

Reply: Thank you for your suggestion.

(1) In the revised version, we have introduced Section 5.2 "Advancements in this Study" to provide a clearer discussion of the method's advantages. We have improved the logical coherence and academic professionalism of the content in Discussion to enhance readability.
(2) Furthermore, we have included a comparative analysis between our research findings and existing methodologies, which better demonstrates the superiority of our approach in terms of accuracy, computational efficiency, and large-scale applicability.

Below is the revised content:

*5.2 Advancements in this study*

*Accurate monitoring of soybean yield is crucial for food policy decision-making and security assessment. While previous studies have primarily explored the impact of environmental factors such as climate on soybean productivity (Guo et al., 2022; Zhao et al., 2023a), few efforts have focused on producing high-resolution soybean yield dataset for China's major soybean-producing regions. To address this gap, our study produced the NortheastChinaSoybeanYield20m dataset, a 20-meter resolution dataset generated through a hybrid framework integrating the mechanistic WOFOST crop growth model and a GRU deep learning algorithm. Unlike purely data-driven approaches that rely on extensive ground data, our approach leveraged both data mining capabilities and mechanistic modelling, which improve the model's interpretability and enhances its potential for transferability across regions. The integration of the WOFOST model ensured the simulation of diverse production scenarios under varying climate, soil, crop variety and management conditions, providing a robust synthetic training data for the GRU network. This combination allowed the model to generate well, even in areas with limited observational data, therefore overcoming common limitations related to data scarcity and high computational costs. Accuracy assessments using both in-situ and statistical yield data confirmed that the generated NortheastChinaSoybeanYield20m dataset delivered reliable yield estimates across field and regional scales (Fig. 5 and 6). The results also verified the model's stability across time and space, reinforcing its potential for large-scale agricultural monitoring and strategic planning.*

*When compared to previous studies using integrated remote sensing data and process-based model to estimate soybean yield, for instance, Baup et al., (2015) reported estimation error ranging from 2% to 18%, our method achieved comparable levels of accuracy. It also outperformed existing field-scale studies (e.g., RMSE = 400.946 kg ha$^{-1}$ in Ren et al., (2023) and MRE of 29.73% in Du et al., (2014)) and*

*municipal-scale models (e.g., RMSE = 16 % in Von Bloh et al., (2023)). Furthermore, the NortheastChinaSoybeanYield20m dataset showed improved performance relative to similar high-resolution soybean yield products from other countries (e.g., annual 30 m soybean yield mapping in Brazil, with $R^2$ values between 0.31 and 0.71 and RMSEs ranging from 275 to 740 kg ha$^{-1}$ (Song et al., 2022).*

*Although studies based on UAV and RGB data have demonstrated even higher soybean yield estimation accuracy (Li et al., 2021, 2024), such methods are often constrained by high costs and limited spatial coverage, making them impractical for large-scale applications. In contrast, the method developed in this study offers a well-balanced solution that combines computational efficiency, high spatial resolution, and strong predictive accuracy. Our approach offers scalable and practical solution for producing high-resolution, large-scale crop yield datasets.*

The conclusion effectively summarizes the study but could be further refined to better highlight the innovation in dataset construction and its practical applications in agricultural management.

Reply: Thank you for your valuable feedback. In the revised version, we have enhanced the conclusion to emphasize the novel aspects of our approach, particularly the integration of the WOFOST model with deep learning, as well as the practical implications of the NortheastChinaSoybeanYield20m dataset for agricultural management.

Here is the revised conclusion.

*This study generated a high-resolution (20 m) soybean yield dataset for Northeast Chinda from 2019 to 2023 (NortheastChinaSoybeanYield20m) using a hybrid framework that couple the WOFOST crop growth model with a Gated Recurrent Unit (GRU) deep learning algorithm. The framework leveraged a comprehensive soybean growth dataset simulated by WOFOST, which accounted for diverse production scenarios, including variations in climates, crop varieties, soil types and agro-managements practices. This approach effectively reduces reliance on ground observation data, which demonstrating enhanced spatiotemporal generalization capabilities.*

*The dataset was conducted using multi-source remote sensing data, with Sentinel-2 derived time-series LAI as the primary input. Yield estimations showed robust performance at both field and municipal scales, achieving RMSE of 287.44 kg ha$^{-1}$ and 272.36 kg ha$^{-1}$, respectively. To address spatial discontinuities in Sentinel-2 data, corrections using MODIS LAI-derived yield maps effectively mitigated seam effects, achieving complementary benefits in temporal and spatial resolution. The final dataset exhibits high temporal stability and spatial continuity, with mean relative errors (MRE) averaging of 11.46 % at the municipal scale and 7.94 % at the provincial scale.*

*The NortheastChinaSoybeanYield20m dataset successfully captures fine-scale*

*spatiotemporal variations in soybean yield, offering potentials for optimizing production strategies, guiding precision agriculture, and enhancing food security and policy.*

---

## Author Comment (AC5)

**Responses to the comments of Editor**

**Article ID:** essd-2024-586
**Title:** NortheastChinaSoybeanYield20m: an annual soybean yield dataset at 20 m in Northeast China from 2019 to 2023
**Authors:** Jingyuan Xu, Xin Du, Taifeng Dong, Qiangzi Li, Yuan Zhang, Hongyan Wang, Jing Xiao, Jiashu Zhang, Yunqi Shen, Yong Dong

Dear Editor,

Thank you very much for your thorough review and valuable comments on our manuscript. Your insightful feedback has significantly contributed to improving the quality and clarity of our work. In response to your suggestions, we have rigorously revised the manuscript. The key modifications include:

(1) **Methodological Refinements:**
- Introduced a comparative analysis between time-series Landsat-derived LAI and stage-averaged LAI at field scale.
- Added a dedicated section on feature selection methodology (Section 5.1).

**(2) Methodological Refinements:**
- Added relevant references to substantiate our approach and enhance the credibility of our methodology, providing stronger support for our research findings.

The detailed point-to-point responses are as follows. Texts in black are the reviewer's comments; those in blue are our responses to the reviewer's comments; and those in *red and italics* are the revised texts appeared in the revised manuscript.

Thanks,

All our best,

Xin Du

The authors developed a deep learning model using a GRU architecture to predict crop yield, utilizing only two predictors: LAImean1 and LAImean2. Given the simplicity of these two predictors, it raises questions about how they can achieve high prediction accuracy. The authors should provide a more detailed explanation of the underlying reasons or mechanisms that enable such effective performance with just these two variables.

**Reply:** Thanks for your suggestion. We have further clarified the reasons behind our use of two predictors ($LAI_{mean1}$ and $LAI_{mean2}$) in the revised manuscript. In addition, we have conducted a field-scale comparative analysis (Section 3.2 and 3.3.2) that directly evaluates the information loss from reducing full LAI time series to two stage-averaged features and the practical trade-offs involved.

**1. Time series vs. two-stages validation**

For each sample plot in 2022 and 2023, we first identified all available Sentinel-2 observation dates and calculated their corresponding Development Stage Values (DVS). We then built two GRU model variants: (a) a full time series GRU that uses DVS-aligned LAI sequences derived from the simulated soybean growth dataset, (b) a simplified GRU that uses two stage-averaged LAI features ($LAI_{mean1}$ (emergence to flowering), $LAI_{mean2}$ (flowering to maturity).

Validation against in-situ yield observations (Fig. 5 and Fig. A4) showed that the time-series GRU model achieved slightly better accuracy (RMSE = 224.81 kg ha⁻¹, MRE = 7.50%), while the stage-averaged model remained competitive (RMSE = 287.44 kg ha⁻¹, MRE = 10.02%). The difference in MRE is around 3% across both years.

**2. Why two Stage-Averaged LAI remain effective**

In the revised manuscript, we have also added a new section in the Discussion (Section 5.1: Selection of model input features) to elaborate on the design and evaluation of candidate predictors.

We systematically evaluated a broad set of candidate predictors (LAI-based, transpiration-based (TRA), and soil moisture-based (SM) and four summary statistics (mean, max, median, cumulative sum) across vegetative, reproductive, and full-season segments (new Section 5.1), following the methodology present by Ren et al. (2023b).

Notably, the results indicate that the two LAI-derived metrics—$LAI_{mean1}$ and $LAI_{mean2}$—outperformed those derived from either a single phenological stage or the entire growing season in terms of correlation with yield. This supports the findings of Ren et al. (2023b) highlighting the complementary value of combining vegetative and reproductive stage indicators. Conceptually, $LAI_{mean1}$ captures vegetative vigor (establishment and biomass accumulation, Kodadinne Narayana et al., (2024)), while $LA_{Imean2}$ reflects reproductive canopy status — together they summarize the two most

yield-informative phases and mitigate the redundancy present in full sequences.

**3. Practical Constraints on Full Time-Series Feature Use**

Although full LAI sequences outperform stage-averaged inputs at local scale, their application at regional scale is constrained by (a) strong spatiotemporal heterogeneity of Sentinel-2 image availability (Fig. A1), which requires constructing a specific time-series input for every site impartial, and (b) resource-intensive in computational and data-management for many different sequence-patterns. The two stage-averaged design is a phenology-informed compromise that preserves most predictive power while ensuring scalability and robustness to data gaps.

**4. Exclusion of TRA and SM Features**

While transpiration and soil moisture are relevant agronomic variables, they were ultimately excluded due to:

- Lack of high-resolution, high-frequency remote sensing products (especially for TRA); and

- Weak or inconsistent correlations with yield in our dataset, possibly due to indirect or stage-specific effects.

In summary, our comparative experiment, expanded feature evaluation, and the discussion on practical limitations demonstrates that (a) the two stage-averaged LAI features are a computationally efficient choice for regional yield mapping, and (b) full time-series inputs offer modest accuracy gains that are best exploited at local scales or in contexts with dense, regular observations. We have added these results and the associated discussion in Section 4.2, Section 5.1, and Figures 5, A4, and A1. We also acknowledged the limitations on the two-stage average LAI in Section 5.4 and discus future improvements (Lines 637 – 642).

We believe these additions and clarifications, supported by relevant literature, strengthen the methodological transparency and scientific rigor of our study. By selecting $LAI_{mean1}$ and $LAI_{mean2}$, we aimed to balance physiological interpretability with practical feasibility, ensuring that the model remains efficient and scalable for regional to continental applications using remotely sensed data.

Thank you again for your valuable feedback. We look forward to your further comments and suggestions.

Below is a part of the revised manuscript for your reference:

**3.2 Development of the Grated Recurrent Unit model (GRU)**

*Trained on the simulated dataset, the GRU constructed based on TensorFlow 2.6 linked simulated environmental inputs to yield outputs. For field scale yield estimation,*

*we first identified all available Sentinel-2 observation dates for each sample plot based on its corresponding Sentinel-2 tiles in 2022 and 2023 (Table 1), and computed the development stage value (DVS) for each date (Section 3.3.1). As LAI is a key biophysical indicator of soybean photosynthetic capacity and productivity (Malone et al., 2002; Shi et al., 2025), we extracted LAI values at the corresponding DVS from the soybean growth dataset to construct DVS-aligned LAI time-series. These DVS-aligned LAI were served as inputs to the GRU model, with simulated yield used as the target variable.*

**Table 1 DOY of available Sentinel-2 images for LAI extraction.**

| Sentinel-2 tiles | DOY |
|---|---|
| *Available Sentinel-2 data in 2022* | |
| 51UYP | 118, 128, 138, 143, 158, 213, 218, 228, 253, 263, 268 |
| 52TCT | 105, 128, 138, 143, 158, 170, 193, 213, 218, 228, 235, 238, 245, 253, 263, 270 |
| 52TDQ | 105, 117, 130, 140, 160, 222, 245, 250, 265, 270 |
| 52TES | 102, 110, 117, 130, 140, 162, 172, 187, 202, 207, 220, 240, 245, 252, 257, 262, 270 |
| 52TFR | 104, 117, 129, 139, 159, 167, 187, 222, 229, 252, 257, 262, 269 |
| 52TGS | 107, 114, 119, 129, 161, 187, 222, 229, 232, 252, 257, 262, 269 |
| *Available Sentinel-2 data in 2023* | |
| 51TWN | 98, 103, 108, 113, 118, 121, 126, 138, 143, 148, 153, 163, 193, 211, 218, 226, 231, 248, 253, 263, 268 |
| 51TYM | 98, 103, 113, 123, 128, 138, 143, 148, 193, 218, 248, 253, 258, 263, 273 |
| 52TCT | 98, 103, 125, 130, 138, 143, 173, 183, 193, 218, 235, 245, 250, 255, 263 |
| 52TDP | 102, 122, 127, 142, 165, 170, 187, 230, 245, 250, 255, 272 |
| 52TDQ | 112, 117, 130, 142, 165, 245, 250, 257, 272 |
| 52TDT | 105, 130, 137, 142, 147, 167, 177, 232, 245, 250, 255, 265, 272 |

*However, the spatiotemporal variability of Sentinel-2 image availability across regions posed challenges for regional-scale yield modelling, as constructing separate GRU models for each date combination demands considerable computational and storage resources. Accounting for the computational efficiency of the model in large areas, two stage-averaged LAI include $LAI_{mean1}$ (mean value of LAI during vegetative growth: emergence to flowering) and $LAI_{mean2}$ (mean value of LAI during reproductive growth: flowering to maturity), were calculated as inputs, while simulated yields acted as outputs. Boken and Shaykewich, (2002) has demonstrated the feasibility of estimating crop yield using average features derived from a specific phenological stage or from the entire growing season. To evaluate the effectiveness of the two stage-averaged LAI in soybean yield estimation, we applied the GRU model trained by the two stage-averaged LAI to yield estimation at the field scale for comparison with model based on DVS-aligned LAI time-series, and further extended its application to the regional scale.*

**3.3.2 Model estimations of soybean yield**

*DVS-aligned LAI values derived from available Sentinel-2 data (Table 1) were firstly used as input of the GRU model for yield estimation at field scale. Meanwhile, averaged LAI values during the vegetative ($LAI_{mean1}$) and reproductive ($LAI_{mean2}$) growth stages were calculated and used as model input for estimations at both field and regional scales.*

**4.2 Yield estimation at field scale**

The field scale performance of GRU models using full DVS-aligned LAI and two stage-averaged LAI was validated against in-situ measurement from 2022 and 2023 (Fig. 5). The estimated yields exhibited strong agreement with observed yield, with $R^2 > 0.65$ ($p < 0.01$) in all scenarios. Validation results (Fig. 5 and Fig. A4) showed that the DVS-aligned GRU model achieved slightly better accuracy (RMSE = 224.81 kg ha⁻¹, MRE = 7.50%), while the stage-averaged model remained competitive (RMSE = 287.44 kg ha⁻¹, MRE = 10.02%). The difference in MRE was around 3% across both years, suggesting that the simplified approach using two stage-averaged LAI was a feasible alternative for yield estimation.

[Figure]

Figure 1: Scatterplots between estimated and observed soybean yield for 2022 and 2023. (a) and (b) show results for 2022 and 2023, respectively, using the full DVS-aligned LAI; (c) and (d) show results for 2022 and 2023, respectively, using two stage-averaged LAI. Error-bars represent one standard deviation indicating the uncertainty of yield estimations. The dashed line represents 1:1 line. ** denotes statistical significance at p < 0.01.

**5.1 Selection of model input features**

In this study, two stage-averaged LAI ($LAI_{mean1}$ and $LAI_{mean2}$) were selected as alternative input features to DVS-aligned LAI for soybean yield estimation. Although full LAI sequences yielded higher accuracy at local scale (Fig. 5 and Fig. A4) , their regional application was limited by (a) strong spatiotemporal heterogeneity of Sentinel-2 image availability (Fig. A1), which required constructing a specific time-series input for every site impartial, and (b) resource-intensive in computational and data-management costs associated with model training and maintaining models for many different sequence-patterns. The two stage-averaged LAI features are a computationally efficient solution for regional yield mapping, while full time-series inputs offer modest accuracy gains that are best exploited at local scales or where dense, regular observations are available.

For further analysis, we systematically evaluated a broad set of candidate features derived from LAI, transpiration (TRA), and surface soil moisture (SM) extracted from the simulated soybean growth dataset. To develop a unified model suitable for long-

*term and large-scale soybean yield estimation, we employed statistical summaries of these features rather than time-series features tied to specific image acquisition dates (as done by Du et al., (2025)). For each variable, four statistical descriptors— including mean, maximum, median, and cumulative sum—were calculated separately for the vegetative growth stage, the reproductive growth stage, and the whole growing season, following the approach of Ren et al., (2023b).*

*As shown in Fig. 11, LAI-derived features exhibited consistently strong correlations with simulated yield (r = 0.54 to 0.88), reflecting the role of LAI as a critical proxy for canopy development, light interception, and biomass accumulation (Cao et al., 2025; Shi et al., 2025). The multi-spectral retrieval of LAI therefore effectively characterizes both structure and physiological status of the crop canopy, supporting its dominant predictive power in our feature set.*

*Notably, the two stage-averaged LAI ($LAI_{mean1}$ and $LAI_{mean2}$) exhibited stronger correlations with yield than features derived from either single phenological stage or the entire growing season, which is consistent with Ren et al., (2023b). Conceptually, $LAI_{mean1}$ captures vegetative vigor (establishment and biomass accumulation, Kodadinne Narayana et al., (2024)), while $LAI_{mean2}$ reflects reproductive canopy status. These two features jointly summarize the two most yield-informative phases and mitigate the redundancy present in full sequences. Among the candidate features, mean-based features outperformed maximum, median, and cumulative counterparts. This likely due to their lower sensitivity to extreme values and day-to-day fluctuations, making them a more stable and representative indicator of canopy conditions across the two growth periods.*

*While some TRA-based features (e.g., $TRA_{sum}$) showed relatively high correlation with yield, they were excluded owing to practical constraints. Current TRA retrieval methods primarily rely on thermal-infrared remote sensing, which typically has coarse spatial and temporal resolution (Hou et al., 2018; Zhang, 2003) limiting its utility for high-resolution, regional mapping. Similarly, SM-related features showed weak or inconsistent correlations with yield across growth stages in our simulations, indicating a limited direct influence on soybean yield production under modeled conditions.*

*In summary, to optimize model inputs for efficient, large-scale applications, and to facilitate the generation of soybean yield dataset, the two stage-averaged LAI features ($LAI_{mean1}$ and $LAI_{mean2}$) were selected. This selection balances physiological relevance and temporal specificity with strong predictive performance and practical feasibility, enabling competitive yield estimation using only two interpretable, remotely sensed retrievable predictors.*

[Figure]

*Figure 2: The absolute Pearson correlation coefficients between each candidate feature and simulated soybean yield, grouped by growth stages: (a) vegetative growth stage; (b) reproductive growth stage; (c) vegetative growth stage combined reproductive growth stage and (d) whole growing season, respectively.*

**5.4 Limitations and future developments**

*Third, using the two stage-averaged LAI introduces additional sources of uncertainty in yield estimation. Excessive temporal aggregation inevitably obscures growth dynamics. Different growth trajectories can produce similar stage-based LAI yet correspond to different yields, increasing the risk of non-unique LAI for spatially yield mappings. This simplification also limits the modelling capacity of GRU architectures, which are specifically designed to exploit sequential dependencies in time series inputs. Future work can explore hybrid approaches that combine stage-based summaries with higher-frequency or full-season time series of vegetation indicators to improve both interpretability and yield prediction robustness.*

[Figure]

[Figure]

*Figure A1: Spatial distribution of the number of available Sentinel-2 images per pixel for each year: vegetative growth stage (top) and reproductive growth stage (bottom) (a) and yearly averages for each growth stage with error-bars representing spatial standard deviation across pixels within the study area (b).*

[Figure]

*Figure A2: Comparison between estimated and observed yield (2022 + 2023). (a)shows the estimates using the full DVS-aligned LAI and (b) shows the results using two stage-averaged LAI. The error-bars represent one standard deviation indicating the uncertainty of yield estimations. Dashed lines represent 1:1 line. ** denotes statistical significance at p < 0.01.*

*Boken, V. K. and Shaykewich, C. F.: Improving an operational wheat yield model using phenological phase-based Normalized Difference Vegetation Index, International Journal of Remote Sensing, 23, 4155–4168, https://doi.org/10.1080/014311602320567955, 2002.*

*Cao, H., Zhao, R., Xia, L., Wu, S., and Yang, P.: Trends in crop yield estimation via data assimilation based on multi-interdisciplinary analysis, Field Crops Research, 322, 109745, https://doi.org/10.1016/j.fcr.2025.109745, 2025.*

*Du, X., Zhu, J., Xu, J., Li, Q., Tao, Z., Zhang, Y., Wang, H., and Hu, H.: Remote sensing-based winter wheat yield estimation integrating machine learning and crop growth multi-scenario simulations, International Journal of Digital Earth, 18, 2443470, https://doi.org/10.1080/17538947.2024.2443470, 2025.*

*Hou, M., Tian, F., Zhang, L., Li, S., Du, T., Huang, M., and Yuan, Y.: Estimating Crop Transpiration of Soybean under Different Irrigation Treatments Using Thermal*

Infrared Remote Sensing Imagery, Agronomy, 9, 8, https://doi.org/10.3390/agronomy9010008, 2018.

Kodadinne Narayana, N., Wijewardana, C., Alsajri, F. A., Reddy, K. R., Stetina, S. R., and Bheemanahalli, R.: Resilience of soybean genotypes to drought stress during the early vegetative stage, Sci Rep, 14, https://doi.org/10.1038/s41598-024-67930-w, 2024.

Malone, S., Ames Herbert, D., and Holshouser, D. L.: Relationship Between Leaf Area Index and Yield in Double-Crop and Full-Season Soybean Systems, ec, 95, 945–951, https://doi.org/10.1603/0022-0493-95.5.945, 2002.

Ren, Y., Li, Q., Du, X., Zhang, Y., Wang, H., Shi, G., and Wei, M.: Analysis of Corn Yield Prediction Potential at Various Growth Phases Using a Process-Based Model and Deep Learning, Plants, 12, 446, https://doi.org/10.3390/plants12030446, 2023b.

Shi, B., Guo, L., and Yu, L.: Accurate LAI estimation of soybean plants in the field using deep learning and clustering algorithms, Front. Plant Sci., 15, https://doi.org/10.3389/fpls.2024.1501612, 2025.

Zhang, R.: Determination of regional distribution of crop transpiration and soil water use efficiency using quantitative remote sensing data through inversion, Sci China Ser D, 46, 10, https://doi.org/10.1360/03yd9002, 2003.